# Inverted activity patterns in ventromedial prefrontal cortex during value-guided decision-making in a less-is-more task

Georgios K. Papageorgiou[1,2], Jerome Sallet [1], Marco K. Wittmann [1], Bolton K.H. Chau[1,3], Urs Schüffelgen[1], Mark J. Buckley[1] & Matthew F.S. Rushworth[1]

Ventromedial prefrontal cortex has been linked to choice evaluation and decision-making in humans but understanding the role it plays is complicated by the fact that little is known about the corresponding area of the macaque brain. We recorded activity in macaques using functional magnetic resonance imaging during two very different value-guided decision-making tasks. In both cases ventromedial prefrontal cortex activity reflected subjective choice values during decision-making just as in humans but the relationship between the blood oxygen level-dependent signal and both decision-making and choice value was inverted and opposite to the relationship seen in humans. In order to test whether the ventromedial prefrontal cortex activity related to choice values is important for decision-making we conducted an additional lesion experiment; lesions that included the same ventromedial prefrontal cortex region disrupted normal subjective evaluation of choices during decision-making.

[1] Wellcome Centre for Integrative Neuroimaging (WIN), Department of Experimental Psychology, University of Oxford, OX1 3UD Oxford, UK. [2] McGovern Institute for Brain Research and Department of Brain and Cognitive Sciences, Massachusetts Institute of Technology, Cambridge, MA 02139, USA. [3] Department of Rehabilitation Sciences, The Hong Kong Polytechnic University, Hong Kong, China. Correspondence and requests for materials should be addressed to G.K.P. (email: georgios.k.papageorgiou@gmail.com)

Human ventromedial prefrontal cortex (vmPFC) activity covaries with the value of attended objects and potential choices[1–6]. Moreover, vmPFC activity reflects the key variable that should guide decisions: the difference in value between one choice and another[7–16] or the value of the default choice[17, 18]. A better understanding of various cognitive processes and their relationship with activity patterns recorded with human neuroimaging techniques such as functional magnetic resonance imaging (fMRI) can be gained by using the same fMRI approach in other species, such as macaque monkeys[19, 20]. In macaques, fMRI recording can also be combined with intervention approaches such as lesions to establish the causal importance of the brain area for a cognitive process. We attempt to do the same here for the case of the vmPFC and value-guided decision-making.

It should be possible to clarify the nature of the contribution vmPFC makes to representation of reward and value and to decision-making by examining the activity of neurons in the homologous area in animal models or by examining the effect of circumscribed lesions. There has, however, been uncertainty about the identity of the human vmPFC region in which activity reflects choice value and its correspondence to brain areas in other primates[21]. In macaque neural activity in an adjacent and partially overlapping region, orbitofrontal cortex (OFC), is more protracted when it is difficult to identify the better of two options because they are close in value[22] but few investigations of more medial areas, medial OFC, or vmPFC have been conducted. Moreover, surprisingly lesions in the same region do not impair decisions between rewarded and unrewarded stimuli[23, 24].

To understand the nature of vmPFC/OFC activity and its relation to decision-making we carried out a series of experiments in macaques. We based the experimental design and analysis on human studies that have focused on activity related to a key decision variable: the difference in value between the choice taken and the choice rejected during a decision[7–16]. When this analysis approach is taken, activity is consistently found in an arc-shaped part of human vmPFC corresponding to the region Mackey and Petrides identify as area 14[25–27]. In human fMRI experiments great care has been taken to show that vmPFC activity is correlated with the subjective value of the choices being considered. Therefore, in experiment 1, we devised a novel behavioral paradigm that allowed separation of the subjective value of choices from the objective amount of reward with which they were associated. In addition, in the same experiment, we show that lesions that include the vmPFC/OFC area disrupt performance of this type of value-guided decision. A variant of this new paradigm is then studied with fMRI in experiment 2. Finally, in experiment 3 we used a different task in which the options' reward values drifted over time and which included task features resembling those of human neuroimaging experiments.

## Results

**Less-is-more effect.** Four control macaques learned associations between three arbitrary visual conditioned stimuli (CS+) and reward outcomes (Fig. 1a): a highly valued (HV) fruit, a less valued (LV) but still rewarding vegetable, or a compound comprising both (CV: CV contained the sum of HV and LV). After macaques reliably discriminated CS+s associated with HV, LV, or CV outcomes from CS−s with no-reward association, they were given choices between various pairings of the three CS+s (Supplementary Figs. 1 and 2). The animals' choice patterns indicate their subjective evaluation and ranking of the CS+s.

When given choices between actual food items control macaques preferred HV to LV foods. Similarly, they preferred HV-stimuli to LV-stimuli (97.5 ± 4% HV choices; one-sample t-test against 50%

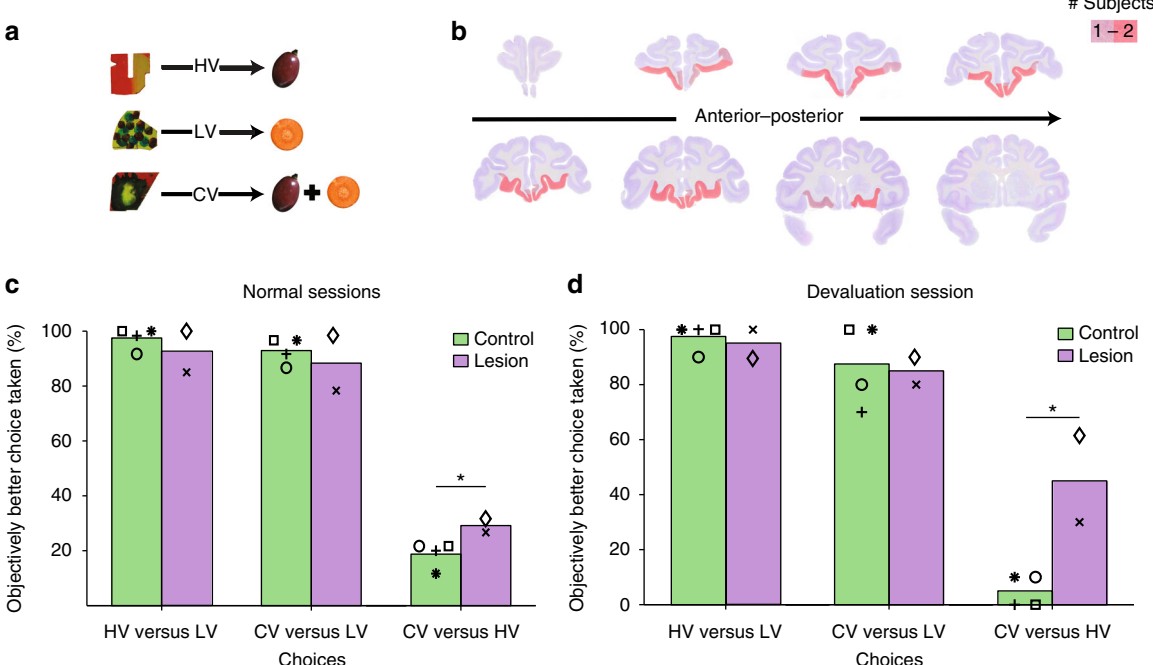

**Fig. 1** Behavioral data—experiment 1. **a** Example stimulus-reward associations for HV, LV, and CV options. **b** Red shading indicates area of vmPFC/OFC lesion present in one or both animals. **c** Frequency of choosing objectively better of the two options. Controls (green bars) as well as lesioned animals (purple bars) preferred HV- to LV- stimuli and CV- to LV-stimuli. For the HV vs. CV decision, strikingly, controls preferred HV-stimuli, receiving only a subset of the rewards that CV-stimuli would have offered. However, macaques with vmPFC/OFC lesions did not prefer CV- to HV-stimuli as much as controls (right). **d** The pattern of results was replicated, including group difference in HV vs. CV decisions, even after HV vs. CV decisions were made easier by satiating animals with LV rewards prior to testing

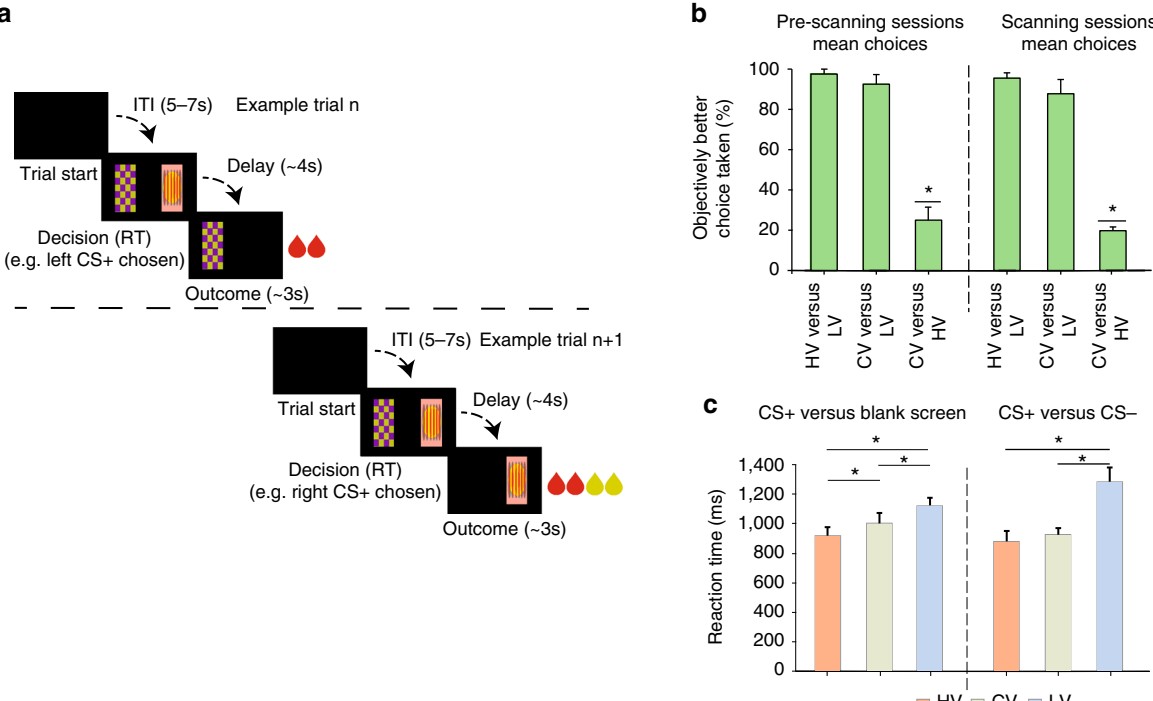

**Fig. 2** Behavioral data—experiment 2. **a** Example trials from fMRI experiment. After an inter-trial interval (ITI) visual stimuli, associated with different outcomes, are presented. Choices were followed, after a mean 4 s delay, with either the delivery of two juice drops (either LV or HV), four juice drops (CV comprising both LV and HV), or no reward. On each trial animals either chose between two stimuli (two-option trials) each associated with reward, in the example illustrated one stimulus is associated with a single reward (left-side stimulus) and the other with a compound reward (right-side stimulus) or, on some trials (one-option trials), a single stimulus was presented on one side of the screen and an animal could either choose it by touching the button placed in front of it or reject it by touching the button in front of the blank side of the screen. **b** Frequency of choosing the objectively better of the two options. Animals preferred HV- to LV- stimuli and CV- to LV- stimuli but they did not prefer CV- to HV-stimuli. **c** Reaction times (RTs) of choices between a CS+ (i.e., HV, CV, LV) and an unrewarded stimulus (the blank side of the screen or a CS−). RTs decreased with expected value of the reward

performance: $t_3 = 23.718$, $p < 0.0005$; Fig. 1c) and CV-stimuli to LV-stimuli (92.5 ± 4.6% CV choices; one-sample $t$-test against 50% performance: $t_3 = 18.304$, $p < 0.0005$; Fig. 1c). Strikingly, macaques nearly always preferred HV-stimuli over CV-stimuli (17.76 ± 4.1% CV choices; one-sample $t$-test against 50% performance: $t_3 = -15.709$, $p = 0.001$). This was the case even though macaques took both components parts of the CV outcome. There were therefore significant differences in the frequency with which macaques chose the objectively better option across the three decisions over ten testing days (repeated-measures analysis of variance (ANOVA): $F_{1.063, 3.189} = 385.402$, $p < 0.0005$). The value expectation linked to the CV-stimulus is biased towards the mean of value expectations for HV-stimuli and LV-stimuli. Therefore, the subjective CS+ evaluations can be dissociated from the objective amount of food predicted. The task thus provides a behavioral assay that can be used when examining whether lesions disrupt decisions based on such evaluations and when examining whether vmPFC signals the subjective value of choices (experiment 2). We note that analogous "less-is-more" effects have been reported under some conditions when humans make decisions[28–30].

**VmPFC/OFC lesion effects.** Lesions of vmPFC/OFC in macaques have comparatively little effect on reward-guided visual discrimination in many circumstances[23, 24]. One interpretation of such a pattern of results is that vmPFC/OFC does not play a critical role in value-guided choice; simple reward-guided visual discrimination tasks may be mediated by representations of stimulus-reward association in other brain regions such as peri-rhinal cortex[31] and striatum[32]. However, a task that separates

subjective values from objective reward amounts, such as the one that we have devised, may be affected by vmPFC/OFC lesions.

We therefore next examined whether vmPFC/OFC lesions disrupted performance of a task that separated subjective value of choices from the objective amount of reward with which they were associated. The distinctive pattern of behavior was significantly diminished in two animals when lesions were placed between the rostral sulcus on the medial surface and lateral orbital sulcus so as to include the prefrontal cortex thought to be most similar to human vmPFC[25–27] as well as adjacent more medial parts of OFC (Fig. 1b). The lesion animals made similar choices to controls when deciding between pairs of HV/LV-stimuli and between CV/LV-stimuli. However, the lesion animals did not prefer the HV-stimuli over CV-stimuli to the same degree as controls (Fig. 1c). A three-way ANOVA with a between-subject factor of group (control, lesion) and within-subject factors of testing day (10 days), and decision (HV−LV, CV−LV, and HV−CV decisions) revealed group differences as a function of decision type (group × decision interaction effect: $F_{1.372, 5.489} = 5.921$, $p = 0.049$; after square-root transformation: $F_{1.345, 5.378} = 8.736$, $p = 0.025$; note that the use of Huynh−Feldt correction meant that the degrees of freedom are slightly reduced after square-root transformation) and a significant effect on the way that animals responded to the three different decisions ($F_{1.372, 5.489} = 288.932$, $p < 0.0005$).

To examine the results in more detail we first focused on the initial nine testing days (Fig. 1c). Two-way ANOVAs across the nine testing days focusing on just HV−CV decisions showed a clear lesion effect ($F_{1,4} = 12.947$, $p = 0.023$) but no similar effects were seen when the other two decisions were examined ($F_{1, 4} < 0.957$, $p$

> 0.05). Widely distributed reward signals elsewhere in the brain[33] may be sufficient for distinguishing the options in these unaffected decisions.

On the tenth day, we considered whether the distribution of lesion effects was simply related to variation in the difficulty of the decisions. Decisions are easy when the green/purple bars in Fig. 1c, d are close to 0 or 100%; the same choice is nearly always taken indicating clear differences in choice values. By contrast, decisions are difficult when an option is chosen in 50% of the trials, suggesting choices are close in value[34]. From this perspective, HV vs. CV choices are difficult for controls. The lesion group's decisions are closer to 50% for all choice types but

particularly for HV vs. CV decisions suggesting lesions might only decrease performance as a function of decision difficulty[35]. Therefore, on day 10, we dissociated difficulty and decision type. Macaques were fed to satiety on LV vegetables prior to testing. Animals experienced the reward outcome associated with each stimulus choice during testing because our test was not intended to investigate inferred revaluation of internal representations of reward goals "on the fly" as is usually the case in devaluation experiments[23, 24, 36] (Methods: Reward Devaluation—Experiment 1). The control macaques' preferences for HV-stimuli vs. CV-stimuli were stronger than before ($5 \pm 5.77\%$ CV choices; one-sample $t$-test against 50% performance: $t_3 = -15.588$, $p = 0.001$),

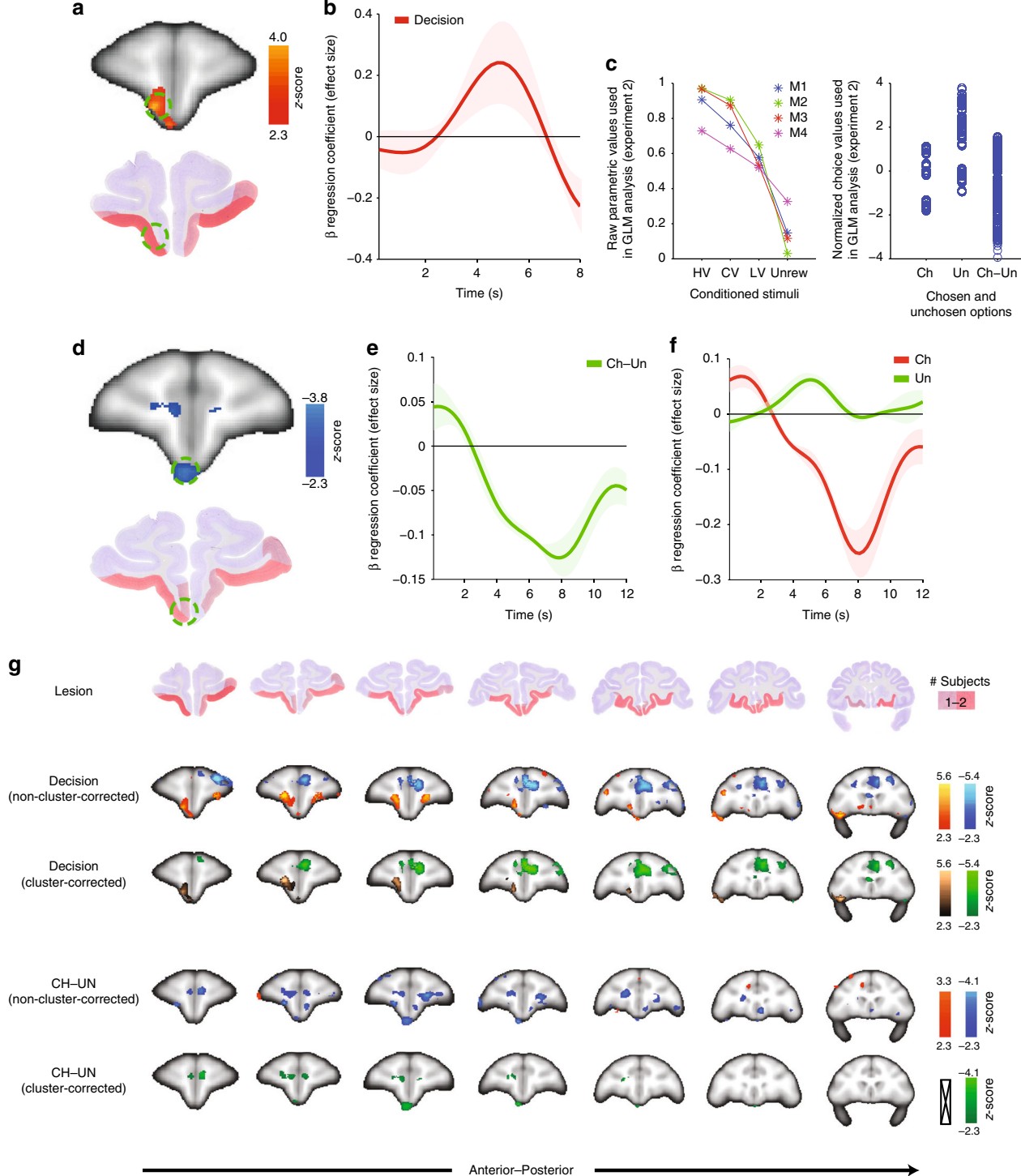

suggesting the CV-stimulus had further decreased in value when animals were satiated on one of its component parts. If the lesion effects were indeed proportional to decision difficulty, then the lesioned animals should also improve (i.e., more closely resemble the unusual pattern exhibited by control animals). However, the lesion-control difference on HV vs. CV decisions was replicated even after reward devaluation (independent-samples $t$-test: $t_4 = -3.939$, $p = 0.017$; Fig. 1d). In conclusion, the lesion disrupted decisions that depended on the animals' subjective evaluation of the stimuli.

**Using fMRI to study value-guided decision-making in macaques**. In experiment 2, a modified version of the task from experiment 1 was performed inside an MRI scanner by the four control animals. The aim was to determine whether activity within the area lesioned in experiment 1 was related to the decision-making and value comparison process. If such activity cannot be found then it suggests that the lesions in experiment 1 may have exerted their effect via an impact on adjacent brain areas[24]. Stimuli were presented on either side of a screen and choices made by pressing one of two infrared sensors nearby (Fig. 2a). Fruit/vegetable outcomes were replaced with juice drops, but the principle remained the same as in experiment 1 (Methods; HV: two drops of high value juice, LV: two drops of low value juice; CV: four drops combining LV and HV). Jittered delays of ~4 s between responses and outcomes allowed dissociation of decision-related neural activity from outcome-related activity using the relatively fast macaque blood oxygen level-dependent (BOLD) signal. Each day's testing comprised 120 trials: 75% one-option trials with one of the three possible CS+s presented and 25% two-option trials with two CS+s presented.

As in experiment 1, macaques preferred HV- to LV-stimuli (Fig. 2b; one-sample $t$-test against 50% performance: $t_3 = 17.301$, $p < 0.0005$) and CV- to LV-stimuli (one-sample $t$-test against 50% performance: $t_3 = 5$, $p = 0.015$) and again exhibited a "less-is-more" effect preferring HV- to CV-stimuli (one-sample $t$-test against 50% performance: $t_3 = -13$, $p = 0.001$). The preferences were also apparent in reaction times (RTs) on single option trials (Fig. 2c). RTs changed with expected reward type (repeated-measures ANOVA: $F_{1.710, 30.787} = 15.348$, $p < 0.0005$); RTs to HV-associated stimuli ($920.5 \pm 117.96$ ms) were faster than to CV-associated stimuli ($1003.87 \pm 133.25$ ms; paired-samples $t$-test: $t_3 = -5.833$, $p = 0.01$) which were, in turn, faster than responses to LV-associated stimuli ($1121.5 \pm 103.36$ ms; paired-samples $t$-test: $t_3 = 3.853$, $p = 0.031$). However, further analysis demonstrated both component parts of CV outcomes had positive values for macaques: all animals learned they could skip a single option trial by touching the sensor in front of the blank side of the screen

when they preferred not to receive the juice. On single option trials HV-, CV-, and LV-associated stimuli were chosen on $91.75 \pm 14.57\%$, $87.63 \pm 14.92\%$, and $77.13 \pm 12.36\%$ of trials, respectively. One-sample $t$-tests against 50% demonstrated all stimuli were chosen more often than they were left (all $t_3 > 4.390$; $p < 0.05$). This shows that all stimuli had positive values and demonstrates that although the CV option had a lower subjective value than the HV option this was not because the LV component within the CV option had a negative, aversive value (Fig. 3c–g, discussed below, presents an alternative analysis leading to a similar conclusion).

These results indicate, more generally, that the animals' RTs resemble those seen during human value-guided decision-making[9]. This was tested further in a complementary analysis that showed that RTs reflected both the overall value of the options available and the difference between the options' values. Across all trials, this was demonstrated by negative effects of both the sum of values of the chosen and unchosen options (one-sample $t$-test: $t_3 = -8.6304$, $p < 0.01$) and the difference between the values of the chosen and unchosen options (one-sample $t$-test: $t_3 = -4.8192$, $p = 0.017$) on RT.

**Increased vmPFC activity at the time of choice**. In order to investigate whether general changes in vmPFC BOLD activity were similar in monkeys compared with humans, we focused our initial fMRI analysis at decision-related events. During these events a large cluster with increased activity was found within the region investigated with lesions (Methods; Supplementary Table 3: GLM-2, contrast 1); it extended from the medial orbital sulcus across the gyrus rectus (possibly areas 14r and 11m[26]; Fig. 3a; cluster-corrected $z > 2.3$; $p < 0.05$). This region has sometimes been referred to as medial OFC (mOFC)[35] but for ease of comparison with humans[25–27], we refer to it as "vmPFC". Just as in human subjects there was a clear effect of decision-making on the BOLD signal. However, while decision-making is accompanied by a decrement in the vmPFC BOLD signal in humans we found that it was linked to a BOLD increment in macaque vmPFC (Fig. 3a, b). Activity positively related to taking a decision was prominent between the medial orbital sulcus and the rostral sulcus in a region corresponding to a large part of the lesion area. Moreover, only activity positively related to decision-making was found within the lesion area (Fig. 3g). The second row and third rows of Fig. 3g show that only positively related yellow/orange and copper-colored activity is found in the area corresponding to the lesion. Activity negatively related to the taking of a decision was confined to frontal regions beyond the OFC and vmPFC such as anterior cingulate cortex and dorsolateral prefrontal cortex (Fig. 3g, second row: blue; third row: green). In humans, decision-

**Fig. 3** VmPFC activity—experiment 2. **a** VmPFC activity increased at time of decision (top; cluster-corrected $z > 2.3$; $p < 0.05$; Supplementary Table 3: GLM-2, contrast 1). Activity increments were prominent in the area removed by the lesion in experiment 1 (bottom). **b** ROI time course illustrating the effect of decision on BOLD activity. Abscissa indicates time from decision onset and ordinate indicates beta regression coefficients relating the decision event to the BOLD signal. Coordinates for ROI correspond to green circle in **a**, top (3, 21, 3). (**c**) Left panel: estimates of each option's value in each individual animal (M1–M4) and **c** Right panel: normalized choice values used in two GLM analyses of experiment 2. However, from left panel **c** it is clear that, prior to normalization, chosen values are usually higher than unchosen values. **d** Activity in vmPFC (1, 17, −2) covaried with the decision variable guiding choices—difference in subjective value (rather than objective reward amount) between choice taken and choice rejected (chosen value-unchosen value; cluster-corrected $z > 2.3$; $p < 0.05$ within 25 mm sphere centered on decision effect in **a**). The regression coefficients relating the BOLD signal to the difference between chosen and unchosen options at the time of choice is plotted in **e** and regression coefficients relating the BOLD signal to the value of the chosen option and the unchosen option are plotted separately in **f**. Difficulty increased, and vmPFC activity increased, when the chosen value was lower or the unchosen value was higher as shown in **e** and **f**. **g** Full summary of lesion location (first row) and of all activity in the frontal lobe positively and negatively related to taking a decision (second row: non-cluster-corrected results; third row: cluster-corrected results) and the decision variable used to guide the decision (chosen–unchosen option value difference; fourth row: non-cluster-corrected results; fifth row: cluster-corrected results). In summary, the results shown in **a** and **d** are representative of the pattern of activity found in adjacent regions and no negative decision-related activity and no positive value-related activity was observed within the lesioned areas

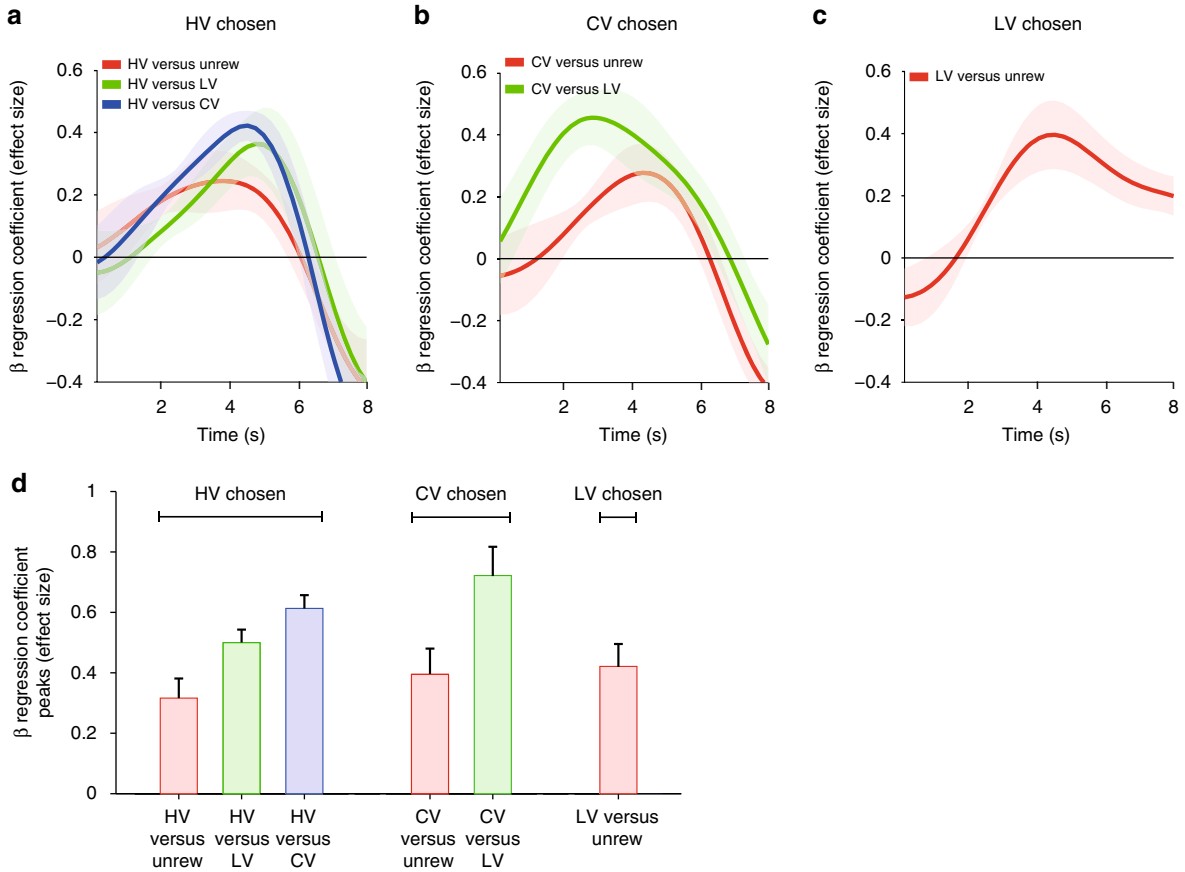

**Fig. 4** Choice-based analysis of vmPFC activity—experiment 2. Time courses of vmPFC activity when HV, CV, LV, and unrewarded options are present (ROI from Fig. 3d). These are the four options whose values are illustrated in Fig. 3c (left panel). Because the impact that the option has on vmPFC activity changes depending on what other option is presented on any given trial (and therefore which option is likely to be taken and which is likely to be rejected) the time courses have been sorted by the value of the chosen option: HV **a**, CV **b**, or LV **c**. The effect of the value of the unchosen option can, for example, be seen in **a**: activity associated with choosing HV is greater when decisions are difficult and choices are made between it and CV (blue) as opposed to LV (green) or Unrewarded (red). The effect of the chosen value can be seen by comparing either the red lines or the green lines in **a** and **b**: activity associated with choosing an option increases when it is harder to make the choice because its value is lower. Although **a–c** show the time courses of the effects of the various options on vmPFC activity, **d** illustrates the same information but using the peaks of the time courses

related activity is associated with activity in a slightly more dorsomedial vmPFC region[2, 3, 8–10], possibly areas 14m and 11m[25–27]. Nevertheless, in both humans and macaques the activity manifests in subdivisions of area 14 and 11m. Neurons with value- and decision-related activity have been found in a more posterior part of this region[37] although recordings of neural activity made this far rostral in vmPFC have not been reported.

**Value-related and decision difficulty-related activity.** Even though decision-making may be associated with activation changes with different signs in humans and macaques, vmPFC activity may reflect value comparison in both species. If this is the case then, as in humans, we should be able to identify activity in macaque vmPFC covarying with the decision variable guiding choices—the difference between the value of the choice taken as opposed to the choice forgone (contrast of chosen value-unchosen value)[8–10]. To pursue this question (Supplementary Table 2 and 3; GLM-2), we first performed an initial analysis in which we contrasted trials when a CS+ was chosen with the small number of trials when the no-reward blank screen was chosen. This analysis allows us to identify activity that is related to choosing any stimulus with any reward association (any CS+) as opposed to any stimulus with no-reward association (any CS−) but it does not reveal whether activity tracks the value of the

choices. We found a relative increase in posterior vmPFC (and OFC activity; Supplementary Fig. 4). Therefore, as suggested by rodent recordings[38], some activity in these regions may not reflect the precise value of a choice but simply whether a choice is guided by stimulus-reward associations.

This initial analysis identified activity linked to the use of stimulus-reward associations *per se* to guide behavior but next we conducted a further analysis focusing only on trials where stimulus-reward associations were being used (we examined just trials on which CS+ s were chosen as opposed to those on which a CS with no-reward association was chosen). In this analysis, we can identify activity that is related to the specific subjective values of the choices that are being considered. By focusing just on trials on which CS+ s were chosen we can ensure that the analysis approach we take does not simply identify activity that is related to using any CS+ as opposed to CS− to guide decisions as in the preceding analysis. To perform such an analysis, we estimated the values that choices held for each macaque by measuring the frequency with which they were taken when offered against the other CS+ s or unrewarded stimuli (Methods: Parametric value analysis—experiment 2; Fig. 3c). This resulted in the choice value estimates for each animal plotted in Fig. 3c (left panel), which were then used to construct regressors coding for the difference in value between choice taken and rejected plotted in Fig. 3c (right panel). Note that normalization was carried out separately on

chosen option values, unchosen option values, and the difference between chosen and unchosen option values. Differences in the distributions of values associated with the options are responsible for making the normalized unchosen value appear higher than the normalized chosen values. These were then used in a parametric GLM analysis (Methods; Supplementary Table 3: GLM-2, contrast 3). We searched for value difference-related activity in cortex within a 25 mm radius of decision-related activity (see Fig. 3a, g). Again, we identified vmPFC activation when brain activity was regressed onto chosen value-unchosen value difference (Fig. 3d; cluster-corrected $z > 2.3$; $p < 0.05$). This result suggests that, as in humans, macaque vmPFC activity reflects the decision variable that should guide behavior: the difference between the value of the options chosen and rejected. However, the relationship between activity and decision variable was negative suggesting activity is higher when it is more difficult to identify the better choice to take. Similarly, in the last two rows of Fig. 3g it is clear that activity related to the decision variable—chosen–unchosen option value difference—is found in several frontal cortical areas but is most prominent within the lesion zone. Moreover, within the lesion zone, activity is only negatively related to the chosen–unchosen value decision variable (Fig. 3g, fourth row: blue; fifth row: green). A positive correlation between activity and chosen–unchosen value difference was only found outside the lesion zone in anterior cingulate cortex (Fig. 3g, fourth row: yellow/red) but this did not survive whole-brain cluster correction (Fig. 3g, fifth row: crossed rectangle).

The relationship between vmPFC activity and chosen–unchosen value was, however, opposite to that seen in humans; as the difference between values decreased (and so decisions became harder), vmPFC activity increased. We took care to search for a value difference effect like that seen in humans but were unable to find one in macaque vmPFC. This remained the case even if we examined smaller volumes of interest surrounding the peak activation effect associated with decision-making or when we examined the region corresponding to the location of the lesion (and adjacent cortex) studied in experiment 1 (Fig. 3g).

This conclusion was supported by further analyses of parametric BOLD activity changes over time (Fig. 3e, f). In a region of interest (ROI), we extracted the raw BOLD time courses, up sampled and aligned them to the decision onset. For each time point and across trials, we calculated the regression coefficient ("effect size") associated with chosen–unchosen value difference from the same GLM (Methods; Supplementary Table 3: GLM-2, contrast 3; Fig. 3e). Next, we illustrated the impact on vmPFC of parametric increases in the chosen value and the unchosen value separately to confirm that they were negative and positive as expected (Fig. 3f). The impact of the value of the selected option is summarized by a single set of regression coefficients (Fig. 3f, red line); vmPFC activity decreases as the chosen option's value increases. By contrast, another single set of regression coefficients illustrates how vmPFC activity increases as the value of the unchosen option increases (Fig. 3f, green line). In combination, these two effects mean that vmPFC activity decreases as the difference between choice values (chosen value −unchosen value) increases (as the decision gets easier; Fig. 4).

In summary, the main effect of taking a decision (activity that changes whenever a decision is taken regardless of the values of the options considered; Fig. 3a, b) is prominent in vmPFC as is activity related to the key variable—the difference in value between the choice taken and the choice rejected—that should drive each decision (Fig. 3d–f). Activity related to the main effect of decision-making and the decision variable is found in partially overlapping voxels at the statistical threshold level we used. The overlap would be slightly more extensive at a lower threshold. In line with this an analysis conducted in a 25 mm radius ROI

centered on the peak effect of decision-making (Fig. 3a, g) revealed a significant effect of the key decision variable—the difference in value between the chosen—unchosen options (Supplementary Fig. 5a).

Additional statistical tests were also used to test the conclusion that vmPFC reflected the key decision variable: the difference in value between the option chosen and rejected. We checked that vmPFC activity still reflected the chosen–unchosen option value difference even when RT (itself also partly determined by the difference in option values) was included in the GLM to explain vmPFC activity. For this analysis, we used appropriate leave-one-out techniques for both selecting the ROIs and determining the time course peaks (one-sample $t$-test: $t_3 = -7.5868$, $p = 0.0048$).

One possibility is that the vmPFC activity pattern reflects some unusual feature of the CV option in which the subjective value was dissociated from the objective value. To test whether this is the case we took three additional measures. Most importantly, we carried out an additional experiment (experiment 3) described below that eschewed CV options. Second, we carried out additional analyses identical to those in Fig. 3e and f but only using data from trials on which the CV option had not been available. We found the same pattern of activity (Supplementary Fig. 5b).

Another way to check that the interpretation of the vmPFC activity identified by the various parametric GLM analyses described above is not unduly affected by the presence of the CV option is to examine activity related to the presence of each of the choice options (HV, CV, LV, and Unrew) in a complementary analysis (Methods; Supplementary Table 3: GLM-1, contrasts based on all cue-onset regressors and sorted by choice taken; Fig. 4a–c). Because the analyses shown in Fig. 3d–f already suggest that the manner in which the presence of the HV, CV, LV, or Unrew option affects vmPFC activity depends on whether or not it is chosen we cannot look simply at trials containing the HV, CV, LV, or Unrew options; in addition, we must consider the context in which each option was presented (what was the other option presented). This complementary analysis had the advantage that it did not depend on precisely how values were assigned to each choice because we simply looked at activity on each of the main decision types. To perform the analysis, we took the peak activity from each trial within a 5 s period between 1.5 s and 6.5 s after stimulus presentation (Fig. 4d). It revealed that vmPFC activity reflected the subjective values, rather than objective reward amounts associated with the stimuli. This was true regardless of whether the options were simple options such as HV and LV or compound options such as CV. Moreover, vmPFC activity increased as the subjective value of the chosen option decreased (compare green and red lines in panels moving left to right; Fig. 4a: HV choices; Fig. 4b: CV choices) and increased as the subjective value of the unchosen option increased (compare red, green, and blue lines; Fig. 4a: HV choices; 4b: CV choices); in a two (chosen value: CV, HV) by two (unchosen value: Unrewarded, LV) factorial ANOVA there was an effect of chosen value ($F_{1, 3} = 15.236$, $p = 0.03$; after square-root transformation: $F_{1, 3} = 36.482$, $p = 0.009$) and unchosen value ($F_{1, 3} = 10.375$, $p = 0.049$; after square-root transformation: $F_{1, 3} = 21.740$, $p = 0.019$). Such a pattern suggests greater aggregate vmPFC activity when identifying the better option was difficult because it had a low value or because the alternative option had a high value and inspection of the CV-related activation patterns confirms that it is the subjective value of choices, rather than objective reward amount that is correlated with vmPFC activity.

Such a pattern of decision-making behavior and vmPFC activity is consistent with decision-making being mediated by a competition between different pools of neurons that reflect the value of two available options[22]. The difference in value between

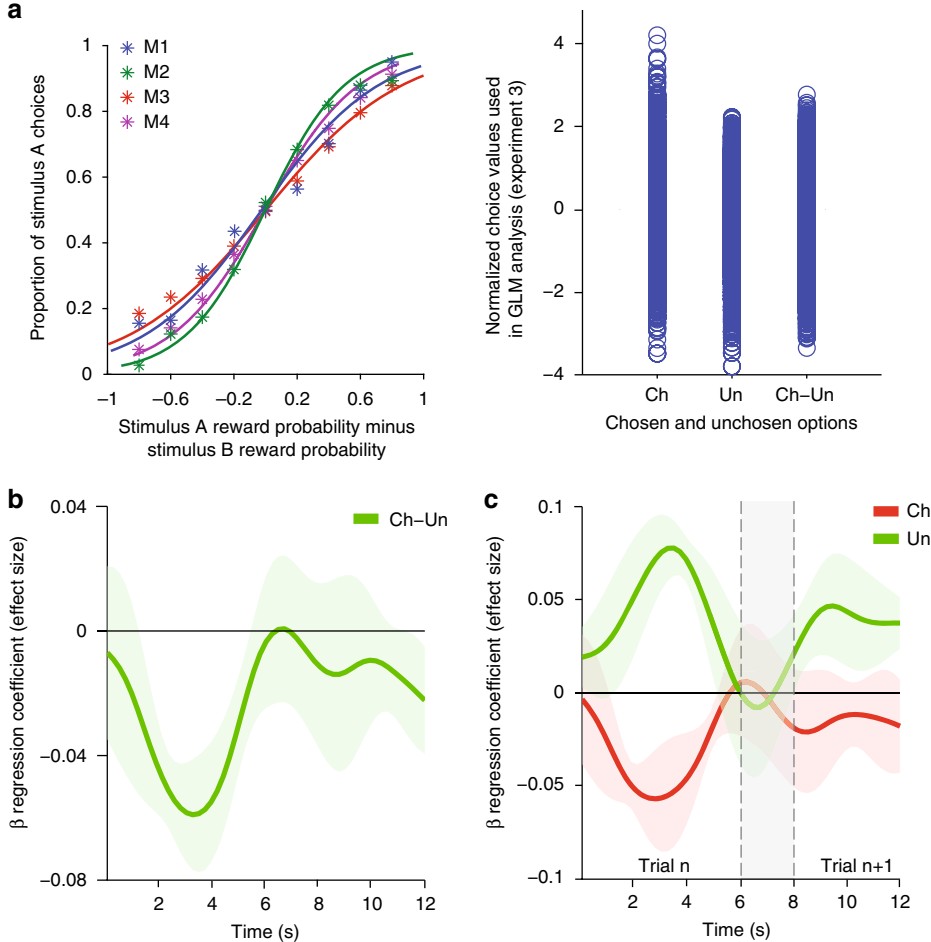

**Fig. 5** VmPFC activity—experiment 3. In experiment 3, each of the three options was associated with a drifting probability of reward. **a** Left panel: sigmoid functions illustrating the proportion of trials on which a stimulus (stimulus A) was chosen as a function of the difference between the values estimated for that stimulus and the alternative option (stimulus B). The value estimates were derived from a standard reinforcement learning algorithm (METHODS: Reinforcement learning—experiment 3). **a** Right panel: normalized choice values used in the GLM analysis. As in Fig. 3c (right panel) normalization was carried out separately on the chosen option value estimates, unchosen option value estimates, and the chosen--unchosen value difference estimates. The effect shown in **b** is unpacked in **c**, demonstrating that activity in macaque vmPFC decreases as the value of the chosen option increases and increases as the value of the unchosen option increases. Note that in experiment 3 trials were performed quickly so that activity in the first seven seconds, approximately, reflects the current trial (trial n). Later activity reflects decisions on subsequent trials (trial n + 1). The gray vertical bar indicates the approximate boundary between trial n and n + 1. On 66% of occasions the option chosen on trial n would be offered again on trial n + 1 (and it was often chosen again) and on 66% of occasions the unchosen option on trial n would be offered again (in which case it was frequently unchosen again) and so the contrasts for trial n capture activity also on trial n+1

two options affects the temporal dynamics with which decision processes can take place[39]. The observed patterns of aggregate BOLD activity are therefore consistent with the neural dynamics of cell assemblies moving back and forth between different states towards a choice[22].

Finally, we note a further analysis of the decisions in experiment 2. We contrasted the decisions that were particularly affected by the vmPFC/OFC lesions in experiment 1 as opposed to the decisions that were less affected by the lesion (GLM-1; Supplementary Fig. 8).

**Replicating value comparison-related activity in macaque vmPFC.** Although our macaque fMRI results can be linked with macaque neurophysiology, the pattern of macaque fMRI activity is surprising because choice value and decision difficulty effects are opposite in humans and macaques (compare panels in Supplementary Fig. 6 summarizing human data with monkey data in the same format); as noted in the Introduction, human vmPFC

activity is positively related to the value difference between chosen and unchosen options and therefore negatively related to decision difficulty[9, 10, 12, 34, 39] (Supplementary Fig. 6a, d, g).

We therefore tested whether we could replicate our findings in experiment 3—a three-option probabilistic reward learning task (Fig. 5a). We have previously used this task to examine activity related to win-stay/lose-switch behavior in a single ROI in posterior lateral OFC[40]. Now we used it to examine choice value-related activity in vmPFC. In this task choice values were continuously and parametrically varied as the reward probabilities associated with the three options drifted during the course of the experiment in a similar manner to human neuroimaging studies (Supplementary Fig. 6). The advantage of this approach is that, because they drift during the course of each session, choice values are distributed throughout the full parametric range in which the GLM analysis is conducted (Fig. 5a—left panel). The GLM analysis of experiment 3 included the same key term as the GLM analysis of experiment 2: chosen–unchosen value difference. Subjective choice values were estimated using a standard

reinforcement learning algorithm[40, 41] (Methods: Reinforcement learning—experiment 3) and reflected the monkeys' observed choice patterns (Fig. 5a—right panel). We used the same analysis strategy in experiment 3 as we had used in experiment 2; we identified the vmPFC location at which decision-related activity (positively) peaked. We then examined activity related to the chosen–unchosen option value difference at this location (at $x$, $y$, $z$ coordinates −3, 17, 7 close to the region studied in experiment 2). We confirmed the findings in experiment 2; macaque vmPFC activity increased with decision difficulty and the value of the unchosen option (activity increased as the difference between chosen and unchosen value decreased: $t_3 = 11.22$, $p = 0.002$; Fig. 5b, c).

A final advantage of the approach used in experiment 3 is that the difference in value between the chosen and unchosen options (sometimes referred to as the "signed difference") and the difference in value between options (sometimes referred to as the "absolute difference" between the options regardless of the choice ultimately taken) are sufficiently decorrelated that both can be employed within the same GLM. When this is done it is clear that chosen–unchosen value difference has a significant impact on vmPFC activity (one-sample $t$-test: $t_3 = −3.155$, $p = 0.004$) but the absolute value does not (one-sample $t$-test: $t_3 = 0.930$, $p = 0.362$; Supplementary Fig. 7). Such a pattern of activity suggests that vmPFC activity is intimately related to the guidance of behavior and/or the current focus of attention[7]. High temporal resolution recording studies in humans[9] and macaques[22] suggest that there is a transition between vmPFC/OFC activity reflecting relative evidence in favor of one-option rather than another during the decision period to activity just reflecting the value of the choice that is taken by the end of the decision.

## Discussion

The results from our two fMRI experiments suggest macaque vmPFC activity occurs when a value-guided decision is taken (Fig. 3a, b) and reflects a key decision variable: the subjective difference in value between a choice taken and a choice rejected (Figs. 3d–f, 4 and 5b, c). By using a compound option, CV, we were able to dissociate subjective value from the objective amount of reward. However, vmPFC value signals did not depend on the presence of such choice options; they were present even when analyses focused on other simpler choice options. In summary vmPFC activity reflects subjective value even when values need not be estimated "on the fly" from "knowledge of the causal structure of the environment"[36]. Activity in this region not only reflects subjective value but in addition it reflects a comparison process that is associated with greater and more protracted activity when the decision is difficult because the choices that are considered are close in subjective value. Therefore, fMRI value difference signals, albeit opposite in sign, can be found in both human and monkey vmPFC.

Although the sign of activity changes is given great weight when interpreting function in human neuroimaging, not just in vmPFC but in other areas such as dorsal anterior cingulate cortex, whether activity is positively or negatively related to difficulty may simply depend on basic features of the networks mediating decision-making[34]. Several studies[9, 10, 12, 16, 42, 43] have shown that activity in frontal lobe areas such as vmPFC can be captured by a variety of computational models of decision-making, which share several features, such as drift diffusion and biophysical cortical attractor models[39]. Such models predict which variables should affect activity and when such influences should arise but they do not make strong predictions about the sign with which BOLD changes should occur; the sign may reflect features of the network that are not integral to the decision process itself but

which are related to whether choice representations are maintained only until decisions are taken or if they are maintained subsequently.

For example, attractor models[39] contain populations of neurons and each represents one possible choice. During the decision process, the network moves to an attractor state in which just a single population reaches a high-firing state. In such a model the high firing attractor state may or may not decay quickly after a decision is reached and this simple difference may be sufficient to flip the sign of a value difference signal recorded with fMRI (see Supplementary Fig. 9 for further discussion). Simple features of neurons in recurrent networks related to their resting activity levels or to the degree to which activity is maintained in a high firing attractor state could therefore produce the activity patterns seen in monkeys or humans. For example, Wong and Wang[44] have described how a change in a single network parameter, the level of recurrent excitation, can determine whether or not the network can make a decision and, if it can make a decision, whether the representation of the choice is maintained in a high firing attractor state. Such a simple change could determine whether the aggregate activity recorded from such a network resembled the pattern seen in humans or in macaques (Supplementary Fig. 9). It is quite plausible that simple features of neurons in networks might vary across species or that their activity may be modulated differently depending on whether primary or secondary reinforcers are used[45]. Either way, the pattern of results is important because it underlines the need for care in interpreting "task-negative" brain areas in human neuroimaging studies[46]. In addition, the results also suggest a similar need for caution when interpreting the fact that activity in other brain areas in human neuroimaging experiments is task positive and increases with task difficulty[34, 47]. Both types of regions may contain value representations that can guide decision-making.

Our finding that decision-making is associated with increments and decrements of activity in macaques and humans respectively may appear at odds with studies emphasizing similarities in default mode activity in the two species[48–50]. We believe that such an emphasis is correct because it is indeed true that there is broadly similar activity in the brains of both species when subjects are at rest. Nevertheless, close inspection suggests that there are differences in the frontal lobe; while default mode activity appears in human vmPFC, the nearest default mode activity in macaques is closer to anterior cingulate cortex (red circles in Supplementary Fig. 10).

Several distinct areas in vmPFC and adjacent perigenual cingulate cortex contribute to valuation and decision-making and there may also be differences in the relative contribution of different areas in humans and macaques[27, 51]. However, the finding that similar information is encoded in fMRI-recorded activity in human and macaque vmPFC, albeit with different signs, suggests that areas with similar cytoarchitecture and similar patterns of interaction with other brain areas[25–27] have a related function involving decision-making and evaluation in the two species. More broadly, such results underline the importance of neurophysiological and other data from animal models for interpreting the data generated by one of the few techniques that can record from the healthy human brain: fMRI. Experiments with fMRI in animal models make it possible to link the two approaches[19, 20].

The lesion results (Fig. 1) suggest vmPFC activity is essential for decisions guided by subjective value estimates. The lesions we studied changed the way that animals chose between HV and CV options but did not disrupt the ability to distinguish the HV from the LV option. This is consistent with observations that lesions in this region have little impact on the ability to learn and use simple stimulus-reward associations and to learn which option is rewarded[52]. Instead such simple stimulus-reward associations

may depend on other brain regions such as the striatum[32] and rhinal cortex[31]. The lesions may have compromised the connections of vmPFC[24] and, in addition to areas 11m and 14, the lesions included adjacent areas 11 and 13 in the central orbital region between the medial and lateral orbital sulci. The fMRI results, however, enable identification of regions within the lesion zone that are closely related to the task; they emphasize activity in vmPFC areas 14 and 11m. Nevertheless, it is important to note that the statistical thresholding used in establishing fMRI activations as significant is conservative. For example, at lower thresholds activity in our experiments was typically more bilateral although it did not extend beyond the lesion zone. fMRI is particularly sensitive to changes in aggregate activity and it is likely that more widespread activity is associated with the reward-guided decision process throughout adjacent OFC areas[22, 53, 54]. Central OFC areas 11 (as opposed to 11m) and 13 may be most important when it is necessary to estimate values "on the fly" from "knowledge of the causal structure of the environment"[23, 36] while more medial vmPFC may be important for decisions guided by value estimates that are constantly updated from experience (experiment 3) or which are constructed from different component elements (experiments 1 and 2).

Control macaques' choices on HV–CV decisions appear irrational and suggest subjective CV value estimates are not optimal. One possibility is that the monkeys' estimation of the CV option is biased away from the sum of the component parts towards their mean. Similarly, humans sometimes average values of groups of items instead of summing them[29]; collectors paid more for sets of high value baseball cards than for identical sets of high value cards with additional low value cards. In another experiment humans valued a set of dinnerware less even if it were larger if it also contained additional broken items[28]. A related pattern of behavior has been reported, albeit in one macaque, previously studied in a task sharing features with those used in experiments 1 and 2[55]. By contrast, animals with vmPFC/OFC lesions were less apparently irrational. Although this may appear to conflict with a vmPFC role in decision-making[56] it is important to note that the lesion-associated difference in behavior occurs because monkeys become increasingly indifferent between the HV and CV options, whereas control animals have clear but counter-intuitive preferences.

One possibility is that the monkeys preferred the CV option to the HV option because of some aspect of the way in which the outcomes were ordered when they were delivered. This seems unlikely because the two component parts of the CV outcome were delivered simultaneously in experiment 1, whereas in experiment 2, although presented sequentially, the CV outcome order was counterbalanced across trials. Another possibility is that the "less-is more" effect is due to the food or juice becoming distasteful when two types are available in the same outcome. However, the food/juices of the CV option were offered sequentially and they were not mixed. Moreover, the sequential consumption of different types of rewards is not very likely to be problematic; foraging animals consume various types of foods in sequence when they are hungry. In experiment 1 the outcomes were presented separately but simultaneously and the animals decided when, how, and in what order to put the items into their mouths.

The choice pattern of control animals may appear more rational if one remembers that decisions in the real world are often made between several multi-component options in contexts where each component outcome is only probabilistically rather than deterministically linked to the animal's choice. In such a scenario evaluating options in terms of their mean value would rarely be detrimental and possibly even efficient. Moreover if decision-making is seen within the broader context of the

foraging decisions macaques evolved to take[57] then the added value of CV options may be outweighed by the handling costs of the LV component and the cost incurred by failing to move on to other opportunities.

An alternative way of thinking about the biasing of the CV option's value towards the mean of its components might refer to value normalization[58–60]. According to such a view the biasing of the CV option's value towards the mean value of its component parts might occur if the monkey's attentional focus is on the HV component within the CV option but if the HV's value is normalized by the presence of the adjacent LV component. An HV option presented in isolation would not be normalized in the same manner.

## Methods

**Subjects**. Six adult macaque monkeys weighing between 6.2 and 13.2 kg and between 8 and 10 years of age participated in the experiment. Two monkeys had lesions in areas that would typically be referred to as OFC in previously studies of macaques. The most medial part of the lesion, however, probably corresponds to a part of the brain that is typically referred to as vmPFC in experiments with humans. We therefore simply refer to the lesions as vmPFC/OFC lesions but a precise description of the area targeted is given in Surgery and Histology. Lesions were made in experiment 1 and the behavioral performance of the animals with lesions was compared with that of four control monkeys. The four control monkeys then continued to experiment 2 in which they were scanned using fMRI. Animals were group housed and kept on a 12-h light–dark cycle, with access to water 12–16 h on testing days and with free water access on non-testing days. All procedures were conducted under licenses from the United Kingdom (UK) Home Office in accordance with the UK The Animals (Scientific Procedures) Act 1986.

**Surgery and histology**. Before any surgeries, monkeys were treated with a steroidal anti-inflammatory (20 mg/kg methylprednisolone injected intramuscularly (i. m.)) and an antibiotic (8.75 mg/kg amoxicillin, i.m.) a minimum of 12 h prior to surgery so as to reduce the risk of postoperative infection, inflammation or edema. During surgery, extra steroidal supplements were provided at 4–6 h intervals. On the morning of the surgery, monkeys were sedated with ketamine (10 mg/kg, i.m.) and xylazine (0.5 mg/kg, i.m.) and were injected with non-steroidal anti-inflammatory agents (0.2 mg/kg meloxicam), atropine (0.05 mg/kg), and opioid (0.01 mg/ kg buprenorphine). These were provided to reduce secretions and provide analgesia. They were further treated with a histamine H2 receptor antagonist (1 mg/kg ranitidine) for gastric ulceration protection due to the administration of both steroidal and non-steroidal anti-inflammatory treatments. Animals were then moved to the operating theater where they were intubated, switched onto sevoflurane inhalational anesthesia and placed in a head holder. Their head was shaved and cleaned using alcohol and antimicrobial scrub (chlorhexidine). Throughout surgeries, respiration rate, heart rate, body temperature, blood pressure, and expired $CO_2$ were continuously monitored.

The surgeries were carried out under sterile conditions and with the aid of a binocular microscope. In surgeries to make vmPFC/OFC lesions, a midline incision was made, the tissue was retracted in anatomical layers and a bilateral bone flap was removed. All lesions were made by aspiration with a fine-gauge sucker. The lateral limit of the lesion was the lateral orbitofrontal sulcus while the medial limit was the inferior bank of the rostral sulcus dorsal to the gyrus rectus. The anterior limit was an imaginary line between the anterior tips of the lateral orbital sulcus and the medial orbital sulcus and extending onto the medial surface to the vicinity of the anterior tip of the rostral sulcus. The posterior limit was an imaginary line between the posterior tips of the lateral orbital sulcus and medial orbital sulcus and extending onto the medial surface to the vicinity of the posterior tip of the rostral sulcus. Once the lesion was made, the wound was closed in anatomical layers. Non-steroidal anti-inflammatory analgesic (0.2. mg/kg meloxicam, orally) and antibiotic (8.75 mg/kg amoxicillin, orally) were administered for a minimum five days after the procedure.

After the end of behavioral testing, both animals with lesions were anesthetized with sodium pentorbarbitone and perfused with 90% saline and 10% formalin. The brains were then removed and placed in 10% sucrose formalin. The brains were blocked in the coronal plane at the level of the lunate sulcus. Each brain was cut in 50-μm coronal sections. Every tenth section was retained for analysis and stained with Cresyl Violet. Examination of the histology confirmed placement of the lesion. Eight coronal sections through the frontal lobes and eight coronal sections through the temporal lobes are shown in Fig. 1b.

In surgeries to implant MRI compatible head posts (Rogue Research, Mtl, CA, USA) a midline incision was made, the tissue was retracted in anatomical layers and the head post fixed with dental cement and Thomas Recording ceramic screws. After a recovery period of at least 2 months, the animals were trained to perform the task inside the actual MRI scanner under head fixation.

**Behavioral training—experiment 1**. In the initial phase of the experiment all monkeys were separated from their group mates in a smaller part of their home cage and in separate but consecutive days they were given free access either to 30 pieces of fruit or 30 pieces of vegetable. The aim was to find appropriate food items that the monkeys were happy to consume. If the monkeys were happy to consume the food offered to them then the next phase of the experiment commenced. If not, the procedure was repeated with a different food type. The experiments involved choices between stimuli associated with either a piece of fruit (high value option: HV), a piece of vegetable (low value option: LV), or a "Compound" option (CV) comprising the same amounts of both the same fruit and the same vegetable. The fruit option used for all monkeys was a grape but because preferences for vegetables differed between monkeys a variety of vegetables were used with different animals (carrot, sugar snap peas, and cucumber) depending on individual animal preferences (two types of vegetable were used with a given animal). These specific foods were used because they had not often been included as standard parts of the animals' daily diet. All fruit/vegetable pieces were prepared so as to be of the same weight and approximately of the same shape. Two of the monkeys that weighed between 6.2 and 9.1 kg were given food pieces that weighed 4–5 g each and the other four monkeys that weighed between 9.9 and 13.2 kg were given food pieces weighing 7–8 g each.

All monkeys had previous experience learning to discriminate between wooden targets to obtain food rewards. For experiment 1, they were trained on a new set of stimuli consisting of 14 targets in total that differed in shape and color pattern (Supplementary Fig. 1). Each target was assigned different reward contingencies across monkeys (example stimulus-reward contingencies are shown in Fig. 1 and Supplementary Fig. 2). During training the monkey sat inside a transport box (62 × 52 × 45 cm) next to a testing table. The experimenter stood on the opposite side of the testing table and, on each trial, presented the monkey simultaneously with a pair of targets. One target led to reward (CS+) and the other target (CS−) was not associated with any reward (Supplementary Fig. 2).

Different CS+ were used for the HV, LV, and CV options (again counterbalanced across animals) and animals learned about each separately. For example, an animal might learn first that a given CS+ was associated with the LV outcome when that CS+ was offered to the animals together with a CS− that led to no reward. Once the animal reached a criterion level of performance (explained below) the animal moved on to a new learning problem. For example, an animal might next learn about the new CS+ associated with the HV outcome but again animals would learn about the new stimulus in the context of trials in which a non-reward stimulus (CS−) was offered. Different CS− stimuli were used in each learning stage. In other words, the CS− stimuli were different when animals learned about the HV CS+, the LV CS+, and the CV CS+. Half of the animals were taught the CS-fruit associations for the HV option first and then the CS-vegetable associations for the LV option, and the other half of the animals were taught in the opposite order. Once a monkey learned about the HV- and LV-associated stimuli, the third pair of targets was introduced so that monkeys could learn about the CV option. If a monkey looked reluctant to take the less valuable component of the CV outcome, the experimenter held it in front of the monkey until it was taken.

The presentation side of the targets (right/left) was assigned pseudorandomly and no target could be presented more than two consecutive times on the same side. The monkey made a decision by touching one of the two targets. In the case of choosing a CS+ target, the experimenter blew on a whistle (to provide an immediate secondary reinforcer), then the unchosen option was withdrawn and the reward associated with the CS+ was offered. Once the monkey took the reward the CS+ target was instantly withdrawn as well. If the monkey chose the CS− target, then both targets were withdrawn, there was no whistling, and an inter-trial interval (ITI) period of ~10 s was starting. The ITI period in all CS+ trials was based on the consumption time.

To minimize the possibility of inadvertent cuing of the monkeys by the experimenter, the experimenter was trained to perform stereotyped movements during testing. He stood behind the testing table so as that the height of his eyes was ~50 cm higher than the height of the monkey's eyes, preventing his gaze from meeting with the monkey's gaze when the latter had to make a choice. The experimenter looked downwards at the center of the testing table and observed the monkey's choices via peripheral vision. Between trials all food rewards were placed on a plastic bowl in front of the experimenter but hidden from the view of the monkey.

Each CS+ learning session during the learning phase consisted of 30 rewarded trials. A monkey needed to perform 30 trials correctly to receive all rewards. If the CS− target was chosen, the trial was repeated until the monkey picked the CS+ target. A criterion was set for a minimum 80% (or 30 out of 37 trials) of trials to be performed correctly within a single session for determining whether a monkey had learned the CS-reward contingencies well. We refer to this initial learning stage as stage 1 (S1).

After learning all the CS-reward contingencies in separate sessions the monkeys proceeded to revision sessions (stage 2, S2) in which they made choices between the HV CS+ and a CS-, the LV CS+ and a CS-, and the CV CS+ and a CS− (in each case the CS− was the one that had been used in the original learning session). When the same accuracy criterion was reached (80% correct performance in a single session) the animal moved onto the main task.

**Behavioral task—experiment 1**. After learning associations between the various CS+ and the HV, LV, and CV outcomes monkeys' choices between CS+s were assessed in stage 3 (S3). The macaques were presented with a choice between:

i.  CS+s associated with the HV and LV rewards,
ii.  CS+s associated with the CV and LV rewards,
iii.  CS+s associated with the HV and CV rewards
iv.  CS+ associated with HV and the CS− used in the same training session, CS+ associated with LV and the CS− used in the same training session, CS+ associated with CV and the CS− used in the same training session. As in the earlier sessions, if a monkey chose the CS− target over its CS+ pair, the trial was repeated until the monkey corrected its error (but no similar procedure was used when animals made choices between two CS+ s).

A revision session (S4) was also given in between the first 2 days of the main task (S3) and the third day of testing on the main task (S5).

After the third day on the main task, a different vegetable was introduced, and the monkeys had to learn the new stimulus-reward associations for new CV and LV options during a new learning phase (S6) that were conducted in a similar manner to S1. These were intended to allow assessment of the generality of the effects that were seen. As before, once the monkeys learned these associations, revision sessions (S7) were given and once the learning accuracy criterion was met, the monkeys moved into the next 3 days of the main task (S8). After the sixth day, revision sessions (S9) were given for the LV and CV stimuli-reward pairings that had been used in the first three sessions (S3, S5) of the main task, and once the learning accuracy criterion was again met the monkeys moved into the last 3 days of the main task (S10). These testing days included all the stimuli and reward types (we refer to the two sets of stimuli learned by the animals as Set A and Set B) learned during the previous days and presented under all possible combinations. The full set of stimuli therefore comprised one HV option (grape), two LV options (two different vegetables), and two CV options (grape combined with the first vegetable and grape combined with second vegetable). Supplementary Table 1 provides a schematic representation of the different stages of the experiment.

**Reward devaluation—experiment 1**. After completing the last 3 days of the main task, all monkeys were given revision sessions (S11) that included only the HV option, one of the two LV options, and the corresponding CV option. Once animals reached the learning accuracy criterion they continued with the Devaluation phase (S12).

Devaluation procedures are often used in the context of investigations of goal-based decision-making and the role of the OFC[24, 36]. In such procedures animals are allowed to feed to satiety on one food type so that its value to the animal decreases. In other variants on the procedure, the food item is devalued by pairing with nausea. Critically, when devaluation is used to assess goal-based decision-making, it is imperative that animals make choices between CS+ s associated with each food type without experiencing the food itself subsequently; this is necessary to ensure that the animals are making choices on the basis of internal representations of the expected outcomes that have been revalued in the absence of direct experience of the particular choice and the outcome in the new sated state. To ensure that this is the case studies with rodents often examine decision-making during extinction when rewards are not actually delivered to the animals[36]. Studies of goal-based decision-making in macaques avoid giving macaques repeated experience of a given CS+ choice in the context of the food type by training the monkeys on multiple pairs of stimuli and ensuring that any given CS+ is only experienced once during the critical decision-making test when CS+ s associated with each of the two foods are paired against one another[24].

By contrast, here the aim of the devaluation procedure was quite distinct. It was not intended to investigate goal-based decision-making on the basis of values inferred "on the fly". Instead the aim was quite simply to decrease the value of the vegetable option. We were interested in investigating the possibility that the CV option, which was partly comprised of the vegetable option, would become even less valuable than the HV option than had previously been the case. The devaluation session was conducted ~24 h after the previous feeding opportunity (so that more than one food type was not inadvertently devalued). A food box (22 × 10 × 14.5 cm) was placed in the monkeys' home cage filled with 250 g of the vegetable option that was to be devalued. The monkey was free to consume the food for 25 min without being directly observed. The experimenter then entered the room and if most of the food was consumed, an additional 150 g of the same food was added. After five minutes the experimenter started observing the animal through the monkey's housing room window until the monkey refrained from consuming any food for 5 mins. The food box was then removed from the home cage and the remaining food was weighed at the end of the session. For all monkeys, 35 min were sufficient to complete this procedure. The monkey was then taken from its home cage and moved into the testing room in a transport box. The main phase of the testing started within 5 mins from the completion of the selective satiation procedure.

During this phase, a task similar to the one described in Main task was given. Supplementary Table 1 provides a schematic representation of the different stages of the experiment.

**Behavioral training—experiment 2.** After experiment 1 was completed the four monkeys without lesions were trained on a modified version of the task used in experiment 1 in which stimuli were presented on a computer screen and which could be performed inside an fMRI scanner. The wooden targets were replaced with clipart pictures displayed on a screen placed ~30 cm in front of the head-fixed animal and the food outcomes were replaced with juice outcomes. A stimulus could be presented on either or both sides of the screen and the monkey could press one of two infrared sensors in front of his left and right hand to make a choice of the spatially adjacent stimulus (Fig. 2).

Six stimuli were used in total: animals learned about three pairs of stimuli where one of the stimuli (CS+s) was associated with the high value (HV), low value (LV), or compound value (CV) reward options. In each case choosing the other stimulus in the pair led to no reward (thus there were three unique CS− stimuli). Stimuli were counterbalanced across animals but the stimulus-reward associations were constant for a given animal across all sessions.

In order to identify appropriate juices for this experiment, all monkeys were separated from their group in a smaller part of their home cage. In the first stage, the experimenter extended a syringe containing 50 ml of a juice toward the animal's mouth and delivered a small volume of juice. A second juice was delivered right after in the same way. In both cases the experimenter carefully observed whether the animal was happy to consume the offered juices. At the second stage, the experimenter gave the animal the opportunity to choose the juice that he most preferred, by extending both syringes at approximately equal distances from the left and right side of the head of the monkey. The presentation side of these syringes was changed every 1–2 trials. The monkey could discriminate the juices not only by their taste and smell but also by the color of the syringes.

Three distinct juices were used: grapefruit, strawberry, and blackcurrant. One juice served as the HV option and a second juice as the LV option (which was used in each depended on the preferences of individual animals). In both cases juice delivery lasted ~500 ms. The CV option consisted of identical amounts of each of the two juices (in other words, each of the two juices that comprised the outcome were delivered for 500 ms separated by a 300 ms interval). The order of the juice delivery in the CV option was counterbalanced so that in the half of the trials juice A was delivered first and then juice B and in the other half the opposite order of presentation was used.

The task consisted of 120 trials, from which 75% (or 90 trials in a given session) were single option trials: only one stimulus was presented on either the left or right side of the screen. The other 25% (or 30 trials) were 'choice' trials: two stimuli were presented on either side of the screen. After extensive training, all monkeys learned that they could skip a given single option trial by touching the sensor that was placed in front of the blank side of the screen. 15% of choice trials consisted of choices between the CS+ and CS− target-pairs and the remaining 15% of the trials consisted of choices between the CS+ targets.

Each trial began with a blank screen (ITI: 5–7 s) and at the end of the ITI one or two stimuli were presented on the screen. During both single option and choice trials, if the CS+ stimulus was chosen, all stimuli disappeared from the screen but ~4 s later the chosen stimulus re-appeared and the juice was delivered (outcome phase: 3 s). Each reward was composed of two 0.5 ml drops of juice delivered by a spout placed near the monkey's mouth. If the blank side of the screen or a CS− stimulus was chosen, during single option and choice trials respectively, then all stimuli disappeared from the screen, and an additional delay of 4 s was added to the ITI period.

**Behavioral training—experiment 3.** Experiment 3 employed a three-option probabilistic reward reversal task in which macaques chose between two stimuli on each trial[40]. Instead of having the same pair of stimuli on every trial, two out of three stimuli were randomly drawn for the animals to choose from (Supplementary Fig. 3a). Each stimulus was associated with a reward probability that changed throughout a session (Supplementary Fig. 3b). Each animal performed five to seven sessions in the MRI scanner. Novel stimuli were used on each day of testing.

Each trial began with a blank screen (inter-trial interval; 5–7 s). Two stimuli were presented on the left and right side (stimuli positions were randomized on every trial) of the screen and subjects had to choose an option by touching one of the two infrared sensors placed in front of their left and right hands that corresponded to the stimuli on the screen. If the correct option was chosen, the unchosen option disappeared and the chosen option remained on the screen and a juice reward was delivered. If the incorrect option was chosen, both stimuli disappeared and no juice was delivered (outcome phase: 1.5 s). Each reward was composed of two 0.6 ml drops of blackcurrant juice delivered by a spout placed near the subject's mouth during testing. Each session lasted for 200 trials.

**Statistics.** Means or medians are reported. Error bars correspond to the standard error of the mean. We used repeated-measures ANOVA for Figs. 1c, 2c and 4a, b, Supplementary Fig. 4, a one-sample t-test for Supplementary Fig. 5a, Fig. 7, a one-sample t-test against 50% for Figs. 1c, d and 2b, an independent-samples t-test for Fig. 1d and paired-samples t-test for Fig. 2c. All statistical tests were two-tailed.

**Imaging data acquisition.** Imaging data were collected using a 3 T MRI scanner and a four-channel phased-array receive coil in conjunction with a radial transmission coil (Windmiller Kolster Scientific, Fresno, CA, USA). FMRI images and reference images for artifact corrections were collected, whereas awake monkeys were head-fixed in a sphinx position in an MRI compatible chair. FMRI data were acquired using a gradient-echo T2* echo planar imaging (EPI) sequence with $1.5 \times 1.5 \times 1.5$ mm$^3$ resolution, TR = 2.28 s, TE = 30 ms, flip angle = 90°. Proton-density-weighted images using a gradient refocused echo sequence (TR = 10 ms, TE = 2.52 ms, flip angle = 25°) were acquired as reference for body motion artifact correction. T1-weighted MP-RAGE images ($0.5 \times 0.5 \times 0.5$ mm$^3$ resolution, TR = 2500 ms, TE = 4.01 ms) were acquired in separate anesthetized scanning sessions. A similar protocol was described by Sallet and colleagues[61].

**fMRI data preprocessing.** FMRI data were corrected for body motion artifact by an offline-SENSE reconstruction (Offline_SENSE GUI, Windmiller Kolster Scientific) method[62] and by performing independent component analysis using FSL MELODIC. The images were aligned to an EPI reference image slice-by-slice (Align_EPI GUI and Align_Anatomy GUI, Windmiller Kolster Scientific) to account for body motion and then aligned to each monkey's structural volume to account for static field distortion[63]. The aligned data were processed with high-pass temporal filtering (3-dB cutoff of 100 s) and Gaussian spatial smoothing (full-width half-maximum of 3 mm). The data that were already registered to each monkey's structural space were then registered to an independent population-average MRI-based template[64, 65] using affine transformation[66].

**fMRI data analysis.** Whole-brain analysis was conducted using a univariate General Linear Model (GLM) approach with FMRIB's Software Library[67]. We searched for brain regions that exhibited activity at the cue onset and at the reward delivery time. To do this we applied two GLMs to every testing session.

The first GLM (GLM-1) included 24 regressors and the second GLM (GLM-2) included ten regressors (Supplementary Table 2). In, GLM-1 we used nine constant regressors, each time-locked to a condition of interest (HV vs. LV, etc.) as well as a nuisance regressor for discarded choice trials. Two sets of these ten regressors were used in the analysis, one set time-locked to cue onsets and one set time-locked to reward deliveries. Four additional regressors indexed events at the time of response. Two regressors indexed whether responses were made with either the left or the right hand. All these regressors were convolved by the standard hemodynamic response function (HRF: a gamma function with 3 s mean and 1.5 s variation reflecting the standard macaque BOLD HRF[40], which is faster than in humans). Two other regressors indexing left and right responses were not HRF-convolved as part of an additional attempt (alongside the other procedures described above) to capture movement related artifacts.

We noticed that the effect of the chosen value in experiment 2 (although not experiment 3) was protracted and it is possible that this may be related to the reactivation of the representation of the chosen option at the time of outcome delivery[68] and the fact that the decision and outcome separation was longer in experiment 2 than experiment 3. Further analyses using slower HRFs (with a hemodynamic lag of 4 s) failed to identify additional areas of activity.

GLM-2 employed a parametric approach. We modeled trials in which a CS+ as chosen and trials where no CS+ was chosen (instead the blank screen or a CS− were chosen to proceed to the next trial) separately. Cue onset and reward delivery were modeled as two events, resulting in four trial types of interest. Only for the cue-onset regressor and for trials in which a CS+ was chosen, we added two parametric regressors. These parametric modulators comprised of one regressor to index the sum of the values of the options offered to the animals and one to index the difference in value between the choice taken and the choice rejected (respectively these regressors are listed below "Chosen+Unchosen value", "Chosen−Unchosen value"). This resulted in six regressors of interest (four contrast regressors, two parametric regressors). GLM-2, like GLM-1, contained, in addition, four motion-related nuisance regressors capturing the effects of the monkey responding by either a right or a left button press either with or without HRF-convolution. As in GLM-1 these regressors control for motion-related neural activity and for motion-related image artifacts, respectively. Contrasts are listed in Supplementary Table 3. Results shown in Fig. 3a, b, g illustrating decision-related activity, Fig. 3c–g illustrating value-related activity and activity related to the difference between chosen and unchosen values are taken from GLM-2.

All analyses were first conducted at the individual monkey level on at least four to six sessions. Average effects of the GLM across sessions within the same monkey were calculated using a fixed-effects analysis. At the group level, analyses were performed using FMRIB's local analysis of mixed effects stage 1 and 2 Flame 1 + 2[69, 70]. Activations exceeding a threshold of z > 2.3 within the vmPFC/mOFC lesion area of interest (defined on the basis of histology; Fig. 1b) are reported as are activations elsewhere in the brain that surpassed the same threshold after cluster correction (in other words, a standard cluster-based thresholding criteria of of z > 2.3 but with a p < 0.05 cluster correction procedure).

**fMRI time course analysis.** To illustrate activity in vmPFC/mOFC, we placed a ROI over the peak of the vmPFC/mOFC signal at the time that the visual stimuli were presented on the screen and extracted the time course of the BOLD signal

from two-voxel radius spherical masks. In some cases, the time courses are shown for illustration. However, in some cases statistical tests were performed on the time courses; this was only done when a statistical test was orthogonal to the contrast originally used to define the ROI. For any statistical analyses, we extracted the time course of activity from the period corresponding to the full-width-half-maximum of the peak established using a leave-one-out procedure to avoid temporal bias and from a location determined using a leave-one-out procedure to avoid spatial bias. In particular, to avoid temporal bias, the maximum value of the mean time course of beta weights of all except the left-out session was used to identify the respective beta values in the left-out sessions. These beta values from the left-out sessions were then statistically tested. To avoid spatial bias, a similar approach was used in which we identified the spatial activity peaks for each session from all other sessions, except the left-out one.

**Parametric value analysis—experiment 2**. For the purposes of this analysis a single value was assigned to each choice each animal could make: each CS+/− or the blank screen. The values were based on a series of calculations: (1) The percentage of occasions each CS+ was chosen when each animal chose between a CS+ or forgoing the trial by touching the sensor in front of the blank side of the screen; (2) The percentage of occasions that each CS+ was chosen when each animal chose between a CS+ and a CS−; (3) The percentage of occasions that each CS+ was chosen when each animal could choose between two CS+s. In each case the percentages were multiplied by the number of instances of the trials on which the choices were presented. In the final stage of value calculation each stimulus' value for an individual animal was calculated by summing the outputs of the three calculations described above and dividing by the total number of trials on which the choice had been offered. For example, the value of the HV option was calculated as follows:

$$HV = \frac{\begin{array}{c}(HV\ versus\ Blank\ side)*\#trials + (HV\ versus\ CS-)*\#trials \\ + (HV\ versus\ CV)*\#trials + (HV\ versus\ LV)*\#trials\end{array}}{Total\#trials}$$

**Reinforcement learning—experiment 3**. In the three-option probabilistic reward reversal task of Experiment 3, the value of each option was estimated using the Rescorla–Wagner model[41]:

$$V_{t+1,s} = \begin{cases} V_{t,s} + \alpha(r_t - V_{t,s}), & \text{if option s was chosen} \\ V_{t,s}, & \text{if option s was unchosen or not presented} \end{cases}$$

where $V_{t,s}$ and $r_t$ are the value of option $s$ and the choice outcome on trial $t$, respectively. $\alpha$ is a learning rate free parameter.

The stochasticity parameter $T$ was estimated by applying a softmax function that models probabilities of choosing each option:

$$P_{t,s} = \frac{\exp(V_{t,s}/T)}{\sum_{s'=1}^{3} \exp(V_{t,s'}/T)}$$

where $P_{t,s}$ is the probability of choosing option $s$ on trial $t$.

The free parameters $\alpha$ and $T$ from the Rescorla–Wagner model and the softmax function respectively were fitted session-by-session by minimizing the negative log likelihood $L$:

$$L = -\sum_{t=1}^{N} \log(P_{t,c_t})$$

where $N$ is the total number of trials and $c_t$ is monkey's choice on trial $t$.

**Code availability**. Data analyses were conducted in FSL and in MATLAB using scripts available from the corresponding author upon reasonable request.

**Data availability**. The data that support the findings of this study are available from the corresponding author upon reasonable request.

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

## Acknowledgments

Funded by the MRC and Wellcome Trust; G.K.P. received support from the A.G. Leventis Foundation. We thank Rhyanne Dale for her assistance in data collection and Greg Daubney for histology. We thank Miriam Klein-Flügge and Nils Kolling for helpful comments on an earlier draft of this manuscript.

## Author contributions

G.K.P. and M.F.S.R. conceived and designed the study; G.K.P., M.F.S.R., M.K.W., B.K.H. C., and U.S. analyzed the data; G.K.P. and J.S. trained the animals; M.J.B. performed the surgeries; G.K.P. collected the data. All authors contributed to the preparation of the manuscript.

## Additional information

**Competing interests:** The authors declare no competing financial interests.

