## [Peer Review File · Nature Communications]

Reviewers' comments:

Reviewer #1 (Remarks to the Author):

Papageorgiou and colleagues investigate the role of the vmPFC in value-based decision making in macaques (using behavioral and fMRI experiments) and contrasted their findings to those reported previously in human neuroimaging. The authors motivate this work by highlighting that one feature of human vmPFC activity is that it often exhibits a task-negative response profile, which in turn makes interpreting the role of this area in valuation difficult. Here, in a series of three experiments, the authors show that the relationship between primate vmPFC activity and value was inverted and opposite to the relationship seen in humans.

I couldn't agree more that the issue of task-negative responses in human medial prefrontal cortex and the extent to which this region actively participates in valuation is crucial (and regrettably not openly highlighted/discussed in many human fMRI studies).

However, direct comparisons between primates and humans are difficult to make. In this paper I see two potential problems: 1) comparable human data were not obtained at the same time to allow direct comparisons and 2) the human fMRI literature is littered with reports conflating results from different parts/subdivisions of the medial prefrontal cortex that often exhibit different response profiles (an extreme example of this would be the tendency of some authors to use the terms vmPFC and OFC interchangeably, despite these areas being anatomically and functionally distinct).

I therefore suggest that the study focuses mainly on what was done here (i.e. reporting animal work alone) and move the more speculative comparisons with the human work to the discussion.

In addition I have some conceptual and methodological concerns that I would like the authors to try and address.

[1] The "Less is more" effect is somewhat counterintuitive. I would be surprised if humans behaved in the same way if they were presented with the same scenario (which would further complicate the attempted species comparisons). Intuitively, since all three rewards/items used here were positively valenced there is no reason to give up the combined offer of a high and low value item (CV trials) over the high value item alone. How do the authors explain the tendency of the animals to scale the value of CV trials down to the average of HV and LV trials? Could there be another explanation for this behavior (e.g. animals reluctant to mix their food/drinks in one go)?

[2] The lesion effects are not particularly convincing. The low sample size ($N=2$) of the lesioned animals is problematic. More critically, however, the analysis used here (three-way repeated measures ANOVA) is inappropriate. Since there is a group comparison here, the authors should have used a mixed ANOVA instead and treated the group factor as a between subject variable. Judging from the currently reported statistical effects on the group \times decision interaction ($p < 0.049$) I doubt that the new analysis would survive the test. More generally, ANOVAs can be problematic when used on bounded quantities such as % choice. Perhaps a single-trial mixed-effects logistic regression taking into account inter-subject variability might be more appropriate (though it won't help much with the low sample sizes).

[3] Would be good to see reaction time data for the first set of behavioral data. This might shed additional light on the "less is more" effect as well as the lesion results. For example would RTs be comparable on (HV vs LV) and (CV vs HV) as would be suggested by the choice behavior? How about the RTs of the lesioned animals – would they be slower in line with their choice behavior?

[4] Why take into account the unrewarded stimuli when estimating the subjective value of the CS+ choices? In principle only the offers against other rewarding stimuli should matter. Assuming animals are aware of the non-rewarding nature of some stimuli they should never choose them over any of the rewarding ones.

[5] When testing the extent to which the vmPFC encodes the value difference between the alternatives it might be useful to look at the absolute difference between the two alternatives as it is often done in human fMRI studies. The difference between chosen minus unchosen item used here is a signed quantity and it won't correlate perfectly with the absolute difference (i.e. judging from the reported behavior there will be a subset of trials in which animals chose the lower valued item leading to negative differences).

[6] Could you please clarify/discuss how we are meant to interpret the findings in Fig. 3a,b in relationship to those in Fig. 3d-f? There is clearly a partial spatial overlap in vmPFC between decision and choice values, so I am wondering if we should treat these two clusters as roughly the same or as two different subdivisions of the vmPFC with potentially different roles?

[7] Relatedly, would it make sense to test the chosen/unchosen effects on the region identified as being linked to the decision and exhibited a task-positive response profile (as shown in Fig. 3a) rather than running a separate GLM and/or contrasts to derive a new cluster (as done in Fig. 3d)?

[8] The difference in vmPFC activity between chosen and unchosen values in Fig. 3e,f appears to be driven almost exclusively by the chosen values, a finding that might contradict the proposed comparator idea and suggest that this subdivision of the vmPFC encodes mainly the value of the chosen option.

[9] Related to this, I wonder whether the sign flip between vmPFC activity and value difference between primates and humans follows trivially by the fact that in animals, unlike humans, we have an above-baseline (i.e. task positive) vmPFC response (i.e. activity associated with hard trials should be at opposite ends across the two species - highly suppressed in humans and highly activated for animals). It's possible I'm missing something here so it would be good to clarify this point further as it is one of the main messages of the paper.

[10] I am not against catchy titles but the one used here seems too generic and does not convey the main message of the paper clearly. Please consider revising the title.

Reviewer #2 (Remarks to the Author):

In this paper Papageorgiou and colleagues present a very interesting set of results, gained via a combination of behavioural testing, lesions and fMRI in macaques. The authors' aim is to dissect the role of vmPFC in guiding value-based choice, and I'm full of praise for the way in which they have tackled this issue by combining different experimental techniques. The paper is timely: very little work has been done on mOFC/vmPFC and economic choice in non-human primates, in particular using fMRI (this is surprising given the vast number of fMRI studies studying the role of vmPFC during value-based choice in humans). The authors' two key results are striking and counter-intuitive when compared with our existing knowledge of human primates. Firstly, they report that the animals systematically prefer the inferior high-value option (HV) to the superior combined option (CV) that includes the HV together with a low valued option (LV). Secondly, they show that the BOLD signal in the macaques' vmPFC codes chosen value vs unchosen value, reversing the usual coding pattern

isolated in humans.

The paper, however, currently struggles to present these results in a coherent fashion. It feels rather like one is reading two papers combined into one. For example, the title and the abstract focus primarily on the behavioural findings (what the authors term the 'less is more effect'), while the vast part of the discussion focuses on the inverse BOLD pattern for value-coding, with little mention of the behavioural findings. This is not just a stylistic problem of how the findings are presented but it runs more deeply in the way analyses are conceived.

For example, let's assume that the authors want to focus on understanding the computational underpinnings of the 'irrational' behaviour that they isolated. There are a number of questions that one would like to see addressed to understand this phenomenon more deeply. For example, in the HV vs CV, does the LV option act as a decoy when presented in combination with the HV, thereby lowering the value of HV? Note that there is a vast literature on this phenomenon in the field of JDM that the authors do not mention. Is the order in which the options are presented in the CV case important? From a more neuroscientific angle, is this effect driven by normalisation? (see for example the work of Louie, Glimcher or Soltani). Moreover, it is unclear via which mechanism the lesion reduces (but does not abolish) this effect. It is also puzzling why the lesion seems to have little effect on the ability to discriminate the most rewarding option. Is this due to the fact that in this type of task the animals are heavily trained on the association between fractal and reward, making this task more dependent on a striatal circuit and less on OFC (that is usually involved in constructing on-the-fly values)? Can the devaluation experiment shed some light on this? In other words, if this is the central topic of the paper (as the title suggests), I would suggest that the authors look into this more deeply and discuss in more detail the theoretical implications of their findings (especially given the fact that the last author has authored many insightful papers on this topic).

On the other hand, if the goal is to understand whether and why the value-coding is reversed compared to humans (and possibly discuss this in relation to the resting state signal which is not reversed), why not focus on the simpler cases in which chosen and unchosen are constructed using LV vs HV, and in which the behaviour is straightforward and in line with the prediction of classical decision theory? I am worried that during the HVvsCV trials other processes might be going on (see point above). For example, let us assume that the effect reported in Figure 3 (e-f) is mostly driven by the CV_vs_HV trials: one might claim that vmPFC is coding the 'underpinning' unbiased value that should be chosen (i.e. $CV > HV$) in the same fashion to the one shown in human studies, even if then (for reasons that are unclear - see previous paragraph) the subjects chose the 'low value' CV option. I see that the authors have already made some comparison between HV and LV but only on the overall signal (Figure 4); however, I could not find a similar analysis for the chosen vs unchosen. I suspect that the last experiment which uses probabilistic reward might help to clarify this issue, but I would like to see this unpacked more in the manuscript.

In conclusion, I believe that the authors potentially have a strong and in part surprising set of findings in their hands. I believe that these findings might spark great interest in the field, but for this very reason, I strongly recommend them to revise the manuscript to make their message less ambiguous.

Reviewer #3 (Remarks to the Author):

This manuscript describes results from behavioral, lesion and fMRI experiments on macaque monkeys performing a reward-based decision making task. Subjects made choices between cues predicting a high value fruit outcome (HV), a lower value vegetable outcome (LV), and a combination of the fruit and the vegetable ($CV=HV+LV$). As expected, monkeys preferred HV over LV and CV over LV cues.

Strikingly, however, monkeys preferred HV over CV cues, even though CV cues predicted more reward in absolute terms. Lesions to the ventromedial prefrontal cortex (vmPFC) and orbitofrontal cortex partially changed this pattern of behavior, such that lesioned monkeys still prefer HV over CV cues, but to a lesser degree than control animals. Imaging data from a similar task involving fruit and vegetable juice indicated that BOLD signals in the vmPFC encode the difference in value between the chosen and the unchosen cue. However, in contrast to a large number of studies on human vmPFC, BOLD responses were negatively (and not positively) related to this value difference.

This is an extremely interesting and important study. The results will be of broad interest and are likely to stimulate much follow-up comparative research. I have several comments that should be addressed.

1. The behavior is remarkable. The authors convincingly show that monkeys prefer cues predicting single high-value rewards over cues predicting the same high-value reward in combination with an additional low-value outcome. There are two potential explanations for this behavior. Either (as the authors suggest) monkeys average the predicted value of the two cues and base their choice on this average value. Alternatively, the value of the two outcomes, when received together, is lower than the sum of the two rewards when received in isolation (somehow, combining a vegetable and fruit ruins the dish). In this case, monkeys would not average two independent value predictions but instead use the value of the combined reward to guide their choice. This is a subtle but theoretically important distinction. To dissociate between these two possibilities the authors could present choice data from situations in which monkeys choose directly between the fruit vs the fruit & vegetable combination (without cues). I am not sure whether this is what the authors refer to in the sentence on page 4: "This was the case even though macaques took both components parts of the CV outcome." Data on this would be essential.

2. Even more astonishing are the findings from the imaging data. The authors, again convincingly, show that BOLD responses in the macaque vmPFC are negatively related to the value of the chosen option and positively related to the value of the unchosen option. This is in direct opposition to a large number of findings in humans. The authors do not offer a particularly compelling explanation for this discrepancy other than citing that both signatures are generally compatible with neural processes related to value computation. Even though I agree with the assessment that both signatures are in line with value computation, how is it possible that two species that are remarkably similar in many aspects of brain function somehow show the opposite BOLD effects in this task? I think it would be premature to just accept this as an effect of "species differences". I believe the authors should thoroughly explore and discuss alternative explanations for this discrepancy such as differences in task, training, data analysis, etc.

3. The current version of the manuscript touches several vmPFC-related concepts including value coding, choice difficulty, default mode network, and value computation in neural networks. However, throughout, these concepts are sporadically picked up and then dropped again without following through with a line of argumentation. Also, it is unclear how the main finding (value encoding is inverted in monkeys) is related to any or all of these concepts (except value coding). Some conceptual streamlining would improve the manuscript.

4. The authors present a number of different analyses of the imaging data and it is at times difficult to follow what a particular analysis is meant to show or add. I wonder whether there are more direct ways to portray the data. For instance, would it be possible to plot parameter estimates from a GLM with HRF-convolved regressors for the different conditions shown in Figure 3 Ci (HV, CV, LV, Unrew)?

5. Also, the authors use a faster HRF which has previously been used for macaque fMRI data analysis.

However, the figures reveal that many effects are in fact driven by slower responses. In particular, key results focusing on chosen-unchosen value show a (negative) peak around 8 s (Figure 3e and f). How do the authors explain these slow responses and could the choice of the canonical HRF used for the GLM analysis substantially affect any of the voxel-wise results?

Reviewers' comments:

Reviewer #1 (Remarks to the Author):

Papageorgiou and colleagues investigate the role of the vmPFC in value-based decision making in macaques (using behavioral and fMRI experiments) and contrasted their findings to those reported previously in human neuroimaging. The authors motivate this work by highlighting that one feature of human vmPFC activity is that it often exhibits a task-negative response profile, which in turn makes interpreting the role of this area in valuation difficult. Here, in a series of three experiments, the authors show that the relationship between primate vmPFC activity and value was inverted and opposite to the relationship seen in humans.

I couldn't agree more that the issue of task-negative responses in human medial prefrontal cortex and the extent to which this region actively participates in valuation is crucial (and regrettably not openly highlighted/discussed in many human fMRI studies).

We thank the reviewer for their interest in our manuscript and are pleased that the reviewer agrees that the issue of task negative responses is one that needs addressing. We have attempted to deal with each of the points raised by the reviewer as shown below. In this document the reviewer's comments are shown in bold font, our replies are in italics, and text from the manuscript is shown in a standard font with some new areas of text highlighted in red.

However, direct comparisons between primates and humans are difficult to make. In this paper I see two potential problems: 1) comparable human data were not obtained at the same time to allow direct comparisons and 2) the human fMRI literature is littered with reports conflating results from different parts/subdivisions of the medial prefrontal cortex that often exhibit different response profiles (an extreme example of this would be the tendency of some authors to use the terms vmPFC and OFC interchangeably, despite these areas being anatomically and functionally distinct). I therefore suggest that the study focuses mainly on what was done here (i.e. reporting animal work alone) and move the more speculative comparisons with the human work to the discussion.

In our revised manuscript we have followed the reviewer's suggestion and we have moved the discussion of the finer points of human neuroimaging including task-negative activations to the Discussion. Instead, in the Introduction we have followed the reviewer's advice and focused on simply explaining the reasons for conducting an experiment in monkeys with the techniques that we have used.

Moreover, in response to the reviewer's comments, we note that we agree that the majority of studies of reward-guided behaviour in monkeys and in humans tend to focus on different brain areas. Respectively they tend to focus on OFC and vmPFC. It is partly because of this difference that we have undertaken our experiment to see if monkeys show a similar activity pattern and in the same areas as humans. Despite the fact that the data from experiments with the two species were not collected at the exact same time, they were collected with the same technique (fMRI), and analyzed in a very similar way in the same laboratory by the same researchers to allow us to compare the results of the two species, Therefore, in accordance with the reviewer's request we have also tried to make this point much clearer in the Introduction of our revised manuscript.

In addition, we agree with the reviewer that there are many human neuroimaging studies of reward-guided behaviour that probably examine quite distinct cognitive processes and which report quite distinct anatomical regions. However, we think that it is nonetheless worth emphasizing that a number of human fMRI studies exist in which the analysis procedures are consistent and in which the findings are consistent; activity related to a key decision variable, the difference between the value of a choice taken and a choice rejected¹⁻⁸ is found within the approximately arc-shaped region that Mackey and Petrides^{9,10} refer to as area 14. We contend that this finding is robust even when many different ways in which the human subjects are provided with evidence about the value of the options that they are choosing between. The range of studies that have found this activation pattern suggest it is a general feature of activity in this part of area 14. We have tried to clarify this in our completely revised Introduction that now reads:

Human ventromedial prefrontal cortex (vmPFC) activity covaries with the value of attended objects and potential choices¹¹⁻¹⁶. Moreover, vmPFC activity reflects the key variable that should guide decisions: the difference in value between one choice and another¹⁷⁻²⁶ or the value of the default choice^{27,28}. A better understanding of various cognitive processes and their relationship with activity patterns recorded with human neuroimaging techniques such as functional magnetic resonance imaging (fMRI) can be gained by using the same fMRI approach in other species, such as macaque monkeys^{29,30}. In macaques, fMRI recording can also be combined with intervention approaches such as lesions to establish the causal importance of the brain area for a cognitive process. We attempt to do the same here for the case of the vmPFC and value-guided decision making.

It should be possible to clarify the nature of the contribution vmPFC makes to representation of reward and value and to decision-making by examining the activity of neurons in the homologous area in animal models or by examining the effect of circumscribed lesions. There has, however, been uncertainty about the identity of the human vmPFC region in which activity reflects choice value and its correspondence to brain areas in other primates³¹. In macaque neural activity in an adjacent and partially overlapping region, orbitofrontal cortex (OFC), is more protracted when it is difficult to identify the better of two options because they are close in value³² but few investigations of more medial areas, medial OFC or vmPFC have been conducted. Moreover, surprisingly lesions in the same region do not impair decisions between rewarded and unrewarded stimuli^{33,34}.

In order to understand the nature of vmPFC/OFC activity and its relation to decision-making we carried out a series of experiments in macaques. We based the experimental design and analysis on a series of experiments in humans that have studied a diverse range of decisions but which have been consistent in examining activity related to a key decision variable: the difference in value between the choice taken and the choice rejected during a decision¹⁷⁻²⁶. When this analysis approach is taken, activity is consistently found in an arc-shaped part of human vmPFC corresponding to the region Mackey and Petrides identify as area 14³⁵⁻³⁷. In human fMRI experiments great care has been taken to show that vmPFC activity is correlated with the subjective value of the choices being considered. Therefore, in experiment 1, we devised a novel behavioral paradigm that allowed separation of the subjective value of choices from the objective amount of reward with which they were associated. In addition, in the same experiment, we show that lesions that include the vmPFC/OFC area disrupt performance of this type of value-guided decision. A variant of this new paradigm is then studied with fMRI in experiment 2. Finally, in experiment 3 we used a different task in which the options' reward values drifted over time and which included task features resembling those of human neuroimaging experiments. We were able to confirm that the findings from experiment 2 could be replicated using a different reward-guided learning task.

In addition I have some conceptual and methodological concerns that I would like the authors to try and address.

[1] The “Less is more” effect is somewhat counterintuitive. I would be surprised if humans behaved in the same way if they were presented with the same scenario (which would further complicate the attempted species comparisons). Intuitively, since all three rewards/items used here were positively valenced there is no reason to give up the combined offer of a high and low value item (CV trials) over the high value item alone. How do the authors explain the tendency of the animals to scale the value of CV trials down to the average of HV and LV trials? Could there be another explanation for this behavior (e.g. animals reluctant to mix their food/drinks in one go)?

We thank the reviewer for the clear comments. They have guided the way in which we have revised our manuscript. First, we note that a related effect to the one that we describe has been reported in human subjects where it is sometimes referred to as a “less is more effect” or “more or less effect”. In our previous manuscript we referred to a study that had shown this effect in human subjects but we have now expanded the discussion to make this point clearer and we have noted it briefly in the revised Introduction.

The reviewer may be interested to see that a related effect is sometimes even found in human academics (<ftp://repec.iza.org/RePEc/Discussionpaper/dp10752.pdf>). This paper reports that when academics rate other academics they rate them as being better if they have produced a set of high impact papers than if they have produced the same amount of high impact papers and some other less impactful papers.

Second, we have tried to clarify our explanation of the phenomenon. In brief we note that while we may often avoid such scenarios in the laboratory, in many real-life cases choice options have many attributes. In order to make a decision it is necessary to come to an evaluation of all of one option’s attributes and all of the other option’s attributes. In many cases, particularly if time is perceived to be important (and in many natural situations that is the case) biasing the evaluation of an option towards the mean of its component attributes will be an effective way of evaluating it in contrast to other multi-component options.

Third, we note that the less is more effect cannot be attributed to the animals being “reluctant to mix their food/drinks in one go” in any simple way. The food/juices of the CV option are offered sequentially and they are not mixed. Moreover, the sequential consumption of different types of rewards is not very likely to be problematic; foraging animals consume various types of foods in sequence when they are hungry. We hope we have acted in accordance with the reviewer’s wishes in noting these arguments in the revised manuscript. In addition, however, in the revised manuscript we do make clear that the evaluation of the CV option may appear more rational if decision-making is seen within the broader context of the foraging decisions animals evolved to take³⁸. The HV and LV components that comprise the CV each have a value but they also have a handling cost which may increase when the components are offered together as a single outcome. It is possible, perhaps, to think of such an explanation as a slight variant on the reviewer’s suggestion that “animals [may be] reluctant to mix their food/drinks”).

Fourth we have noted explanations put forward by reviewer 2 that focused on 1) the order in which component parts of the CV outcome were delivered (we think that this cannot explain the pattern of results) and 2) which make reference to value normalization effect.

In summary on the revised manuscript we note:

“Control macaques’ choices on HV-CV decisions appear irrational and suggest subjective CV value estimates are not optimal. **One possibility is that the monkeys’ estimation of the CV option is biased away from the sum of the component parts towards their mean.** Similarly, humans sometimes average values of groups of items instead of summing them³⁹; collectors paid more

for sets of high value baseball cards than for identical sets of high value cards with additional low value cards. In another experiment humans valued a set of dinnerware less even if it were larger if it also contained additional broken items⁴⁰. A related pattern of behavior has been reported, albeit in one macaque, previously studied in a task sharing features with those used in experiments 1 and 2⁴¹. By contrast, animals with vmPFC/OFC lesions were less apparently irrational. While this may appear to conflict with a vmPFC role in decision-making⁴² it is important to note that the lesion-associated difference in behavior occurs because monkeys become increasingly indifferent between the HV and CV options while control animals have clear but counterintuitive preferences.

One possibility is that the monkeys preferred the CV option to the HV option because of some aspect of the way in which the outcomes were ordered when they were delivered. This seems unlikely because the two component parts of the CV outcome were delivered simultaneously in experiment 1 while in experiment 2, although presented sequentially, the CV outcome order was counterbalanced across trials. Another possibility is that the “less is more” effect is due to the food or juice becoming distasteful when two types are available in the same outcome. However, the food/juices of the CV option were offered sequentially and they were not mixed. Moreover, the sequential consumption of different types of rewards is not very likely to be problematic; foraging animals consume various types of foods in sequence when they are hungry. In experiment 1 the outcomes were presented separately but simultaneously and the animals decided when, how, and in what order to put the items into their mouths.

The choice pattern of control animals may appear more rational if one remembers that decisions in the real world are often made between several multi-component options in contexts where each component outcome is only probabilistically rather than deterministically linked to the animal’s choice. In such a scenario evaluating options in terms of their mean value would rarely be detrimental and possibly even efficient. Moreover if decision-making is seen within the broader context of the foraging decisions macaques evolved to take⁴³; then the added value of CV options may be outweighed by the handling costs of the LV component and the cost incurred by failing to move on to other opportunities.”

[2] The lesion effects are not particularly convincing. The low sample size (N=2) of the lesioned animals is problematic. More critically, however, the analysis used here (three-way repeated measures ANOVA) is inappropriate. Since there is a group comparison here, the authors should have used a mixed ANOVA instead and treated the group factor as a between subject variable. Judging from the currently reported statistical effects on the group x decision interaction ($p < 0.049$) I doubt that the new analysis would survive the test. More generally, ANOVAs can be problematic when used on bounded quantities such as % choice. Perhaps a single-trial mixed-effects logistic regression taking into account inter-subject variability might be more appropriate (though it won’t help much with the low sample sizes).

Thank you for your comment. However, we note that non-human primate studies typically have similar numbers of subjects. Moreover, we think that there may have been a misunderstanding about the nature of the statistical analysis we ran; we did not perform a three-way ANOVA in the way that the reviewer suggests but rather we performed it in exactly the manner that the reviewer argues it should have been performed – with a between subject factor of group. We hope that a small change in wording will make this clear (we thank the reviewer for spotting this lack of clarity in our original description). The revised manuscript now makes clear that we performed:

“A three-way analysis of variance (ANOVA) with a between-subject factor of group (control, lesion) and within-subject factors of testing day (ten days), and decision (HV-LV, CV-LV, and HV-CV decisions) revealed group differences as a function of decision type...”

The reviewer argues that the statistical strength of the effect at $p=0.049$ is weak. However, we note that the effect appears robust. First the effect is apparent at this level of significance even after Huynh-Feldt correction. Second, we note that bounded quantities can become closer to a Gaussian distribution by square-root transformation. We conducted this and the effect's p value further decreased to 0.025. Third, we note that two-way ANOVAs across the nine testing days focusing on just HV-CV decisions showed a clear lesion effect ($p=0.023$) and a second test incorporating a variant in procedures again revealed an effect of $p=0.017$. In the revised manuscript we have added details of the results after square-root transformation as follows:

A three-way analysis of variance (ANOVA) with a between-subject factor of group (control, lesion) and within-subject factors of testing day (ten days), and decision (HV-LV, CV-LV, and HV-CV decisions) revealed group differences as a function of decision type (group \times decision interaction effect: $F_{1,372,5,489}=5.921$, $p=0.049$; after square-root transformation: $F_{1,345, 5,378}=8.736$, $p=0.025$; note that the use of a Huynh-Feldt correction meant that the degrees of freedom are slightly reduced after square-root transformation.

[3] Would be good to see reaction time data for the first set of behavioral data. This might shed additional light on the “less is more” effect as well as the lesion results. For example would RTs be comparable on (HV vs LV) and (CV vs HV) as would be suggested by the choice behavior? How about the RTs of the lesioned animals – would they be slower in line with their choice behavior?

Unfortunately, we do not have reaction time data for experiment 1; it was not possible to measure reaction times accurately in the set-up that we used. However, we do provide this information for experiment 2. Part of the motivation for changing aspects of procedures in experiments 1 and 2 was to provide precisely this type of data as well as to show the replicability of the effect. We show the reaction times in both the pre-scanning and scanning sessions that were part of experiment 2 (Fig. 2c; left hand panel and right hand panel). The results corroborate the inferences made from the choice pattern (Fig. 2b). Monkeys had faster reaction times to preferred options presented in isolation; the reaction times suggest that the HV option is preferred to the CV option which is, in turn, preferred to the LV option. The figure shows the following:

In addition, the manuscript notes:

“As in experiment 1, macaques preferred HV- to LV-stimuli (Fig. 2b; $t_3=17.301$, $p<0.0005$) and CV- to LV-stimuli ($t_3=5$, $p=0.015$) and again exhibited a “less is more” effect preferring HV- to CV-stimuli ($t_3=-13$, $p=0.001$). The preferences were also apparent in reaction times (RTs) on single option trials (Fig. 2c). RTs changed with expected reward type ($F_{1.710,30.787}=15.348$, $p<0.0005$); RTs to HV-associated stimuli ($920.5\pm117.96\text{ms}$) were faster than to CV-associated stimuli ($1003.87\pm133.25\text{ms}$; $t_3=-5.833$, $p=0.01$) which were, in turn, faster than responses to LV-associated stimuli ($1121.5\pm103.36\text{ms}$; $t_3=3.853$, $p=0.031$). However, further analysis demonstrated both component parts of CV outcomes had positive values for macaques: all animals learned they could skip a single option trial by touching the sensor in front of the blank side of the screen when they preferred not to receive the juice. On single option trials HV-, CV-, and LV-associated stimuli were chosen on $91.75\pm14.57\%$, $87.63\pm14.92\%$, and $77.13\pm12.36\%$ of trials respectively. One sample t-tests against 50% demonstrated all stimuli were chosen more often than they were left (all $t_3>4.390$; $p<0.05$). This shows that all stimuli had positive values and demonstrates that although the CV option had a lower subjective value than the HV option this was not because the LV component within the CV option had a negative, aversive value (Fig. 3c-g, discussed below, presents an alternative analysis leading to a similar conclusion).”

[4] Why take into account the unrewarded stimuli when estimating the subjective value of the CS+ choices? In principle only the offers against other rewarding stimuli should matter. Assuming animals are aware of the non-rewarding nature of some stimuli they should never choose them over any of the rewarding ones.

The reason we take into account the unrewarded stimuli when estimating the subjective value of the CS+ is simply the need to include all the different pairings that a CS+ is contrasted with including those that are unrewarded in order to obtain a quantitative estimate of the difference in value between any all stimuli. The unrewarded stimuli, which were presumably neutral in valence, were also useful for demonstrating that all options were rewarding, even the LV stimuli. Subjective value estimates are then used in the regression against vmPFC BOLD. If we had not used this procedure we would not have been able to estimate the value of the HV, CV, and LV options with respect to unrewarded choices (fourth column) in Fig. 3ci. And without this, we would not be able to estimate the values of all choices taken and rejected (Fig. 3cii).

[5] When testing the extent to which the vmPFC encodes the value difference between the alternatives it might be useful to look at the absolute difference between the two alternatives as it is often done in human fMRI studies. The difference between chosen minus unchosen item used here is a signed quantity and it won't correlate perfectly with the absolute difference (i.e. judging from the reported behavior there will be a subset of trials in which animals chose the lower valued item leading to negative differences).

We have followed the reviewer's suggestion. In experiment 3 the signed difference in value between choice options (we refer to this as "signed difference" from here onwards) and the absolute difference in value between options (referred to as "absolute difference" from here onwards) are sufficiently decorrelated that both can be employed within the same GLM. When this is done it is clear that the signed difference has a significant impact on vmPFC ($t_3=-3.155$, $p=0.004$) but the absolute value does not. We have noted this in new analysis in the revised manuscript as follows:

A final advantage of the approach used in experiment 3 is that the difference in value between the chosen and unchosen options (sometimes referred to as the "signed difference") and the difference in value between options (sometimes referred to as the "absolute difference" between the options regardless of the choice ultimately taken) are sufficiently decorrelated that both can be employed within the same GLM. When this is done it is clear that chosen-unchosen value difference has a significant impact on vmPFC activity ($t_3=-3.155$, $p=0.004$) but the absolute value does not ($t_3=0.930$, $p=0.362$; Fig. S7). Such a pattern of activity suggests that vmPFC activity is intimately related to the guidance of behaviour and/or the current focus of attention¹⁷.

In the supplementary materials we now include the following additional figures, analyses, and legends:

Figure S7. vmPFC activity related to the signed difference (ch-un) and absolute difference(|ch-un|) between chosen and unchosen options. The signed difference in value between the chosen and unchosen option values (shown in brown color) had a significant impact on vmPFC activity ($t_3=-3.155$, $p=0.004$) while the absolute difference in the options' values regardless of choice (shown in blue color) did not ($t_3=0.930$, $p=0.362$). Such a pattern of activity suggests that vmPFC activity is intimately related to the guidance of behavior and/or the current focus of attention.

[6] Could you please clarify/discuss how we are meant to interpret the findings in Fig. 3a,b in relationship to those in Fig. 3d-f? There is clearly a partial spatial overlap in vmPFC between decision and choice values, so I am wondering if we should treat these two clusters as roughly the same or as two different subdivisions of the vmPFC with potentially different roles?

We have attempted to make the relationship between the effects shown in Fig. 3a and b on the one hand and Fig. 3d-f on the other hand. In brief there is some partial overlap in the voxels related to each contrast and this would increase if a more lenient statistical criterion were used. Probably the best way to summarize the result is to say that the main effect of taking a decision (activity that changes whenever a decision is taken regardless of the values of the options considered) is prominent in vmPFC as is activity related to the key variable – the difference in value between the choice taken and the choice rejected – that should drive each decision. Activity related to the main effect of decision-making and the decision variable is found in partially overlapping voxels at the statistical threshold level we used and would be slightly more extensive at a lower threshold. In line with this, when we carried out an analysis suggested by the reviewer in their next point – to analysis the chosen-unchosen value effect at the peak of the decision effect we were able to show that there was a significant effect. The revised manuscript says:

“In summary, the main effect of taking a decision (activity that changes whenever a decision is taken regardless of the values of the options considered; fig.3a, b) is prominent in vmPFC as is activity related to the key variable – the difference in value between the choice taken and the

choice rejected – that should drive each decision (fig.3d-f). Activity related to the main effect of decision-making and the decision variable is found in partially overlapping voxels at the statistical threshold level we used. The overlap would be slightly more extensive at a lower threshold. In line with this an analysis conducted in a 25mm radius region of interest (ROI) centered on the peak effect of decision-making revealed a significant effect of the key decision variable – the difference in value between the chosen-unchosen options (Fig. S5a).”

In the supplementary materials, we now include the following figure and legend:

Figure S5. Related to Figure 3. Control analyses for “inverse” value activity pattern. (a) Value-related activity at the decision-related ROI (GLM-1) and **(b)** in choices in which the CV option is not available (GLM-2), in experiment 2. **(a)** The difference in value between the chosen and unchosen option values (shown in green color) had a significant impact on vmPFC activity (after square root transformation: $t_3=-3.4307$, $p=0.041$) in an ROI around the peak of the main effect of decision-making (Fig. 3a). Such a pattern of activity suggests the main effect of taking a decision (activity that changes whenever a decision is taken regardless of the values of the options considered; shown in Fig. 3a, b) is found in vmPFC in the same regions as activity related to the key variable – the difference in value between the choice taken and the choice rejected – that should drive each decision (shown also in Fig. 3d-f). **(b)** The regression coefficients relating the BOLD signal to the difference between chosen and unchosen options at the time of choice in experiment 2 is plotted in the same manner as in figure 3e but now trials in which the CV option was offered have not been included in the analysis. This means that the chosen-unchosen regression coefficients are based on data from trials on which only the single component HV, LV, and unrewarded options were offered to animals.

[7] Relatedly, would it make sense to test the chosen/unchosen effects on the region identified as being linked to the decision and exhibited a task-positive response profile (as shown in Fig. 3a) rather than running a separate GLM and/or contrasts to derive a new cluster (as done in Fig. 3d)?

As we have noted in response to the previous point, point 6, this is exactly what we have now done in our revised manuscript. We thank the reviewer for this suggestion. Note that the main effect of taking a decision (activity that changes whenever a decision is taken regardless of the

values of the options considered; shown in Fig.3a,b) and activity related to the key decision variable – the difference in value between the choice taken and the choice rejected – are derived from contrasts of regressors used in a single GLM (GLM-2) that is explained in detail in the supplementary materials and in supplementary table S2.

[8] The difference in vmPFC activity between chosen and unchosen values in Fig. 3e,f appears to be driven almost exclusively by the chosen values, a finding that might contradict the proposed comparator idea and suggest that this subdivision of the vmPFC encodes mainly the value of the chosen option.

First, while it is true that the negative chosen value effect has a greater size than the positive unchosen value effect in experiment 2 (Fig. 3f; although the chosen and unchosen value effects were both each significant) the opposite is the case in experiment 3 (Fig. 5c). Second, both chosen and unchosen value effects determine activity levels in both experiments. This can also be seen in Fig. 4 which shows that both the value of the choice taken (generally the higher value option in any decision) and the value of the choice rejected (generally the lower value option in any decision) impact on vmPFC activity.

[9] Related to this, I wonder whether the sign flip between vmPFC activity and value difference between primates and humans follows trivially by the fact that in animals, unlike humans, we have an above-baseline (i.e. task positive) vmPFC response (i.e. activity associated with hard trials should be at opposite ends across the two species - highly suppressed in humans and highly activated for animals). It's possible I'm missing something here so it would be good to clarify this point further as it is one of the main messages of the paper.

In brief our interpretation of the sign flip in vmPFC activity is not unlike the reviewer's and we hope that we have conveyed this in the revised manuscript. We have, however, attempted to make a similar argument to the reviewer's but couched in the terms employed by one of the most commonly used models of a decision-making circuit.

Like the reviewer, our view is that the sign of activity change may be relatively trivial but we note that this view is far from the accepted view. For example, while it is not our main aim to make this point here, one could argue that the idea that dorsal anterior cingulate cortex (dACC) is concerned with conflict or difficulty is a view that is based exclusively on the sign with which evidence for or against a choice taken or rejected impacts on the dACC BOLD signal; it has the same impact as on monkey vmPFC – activity increases with difficulty because it increases with the value of the choice rejected and decreases as the value of the chosen option increases. This has remained the dominant account of ACC activity for twenty years yet we know that it does not accord with lesion and neurophysiological data. We think, therefore, that the field would benefit if more cognitive neuroscientists took the same view as the reviewer and thought of the sign of effects in vmPFC and dACC in neuroimaging studies as more “trivial”. In the revised manuscript we have written as follows:

Although the sign of activity changes is given great weight when interpreting function in human neuroimaging, not just in vmPFC but in other areas such as dorsal anterior cingulate cortex⁴⁴, whether activity is positively or negatively related to difficulty may simply depend on basic features of the networks mediating decision-making⁴⁴. Several studies^{19,20,22,26,44,45} have shown that activity in frontal lobe areas such as vmPFC can be captured by a variety of computational models of decision making, which share several features, such as drift diffusion

and biophysical cortical attractor models⁴⁶. Such models predict which variables should affect activity and when such influences should arise but they do not make strong predictions about the sign with which BOLD changes should occur; the sign may reflect features of the network that are not integral to the decision process itself but which are related to whether choice representations are maintained only until decisions are taken or if they are maintained subsequently.

For example attractor models⁴⁶ contain populations of neurons and each represents one possible choice. During the decision process, the network moves to an attractor state in which just a single population reaches a high-firing state. In such a model the high firing attractor state may or may not decay quickly after a decision is reached and this simple difference may be sufficient to flip the sign of a value difference signal recorded with fMRI (see Fig. S10 for further discussion). **Simple features of neurons in recurrent networks related to their resting activity levels or to the degree to which activity is maintained in a high firing attractor state could therefore produce the activity patterns seen in monkeys or humans. For example Wong and Wang⁴⁷ have described how a changes in a single network parameter, the level of recurrent excitation, can determine whether or not the network can make a decision and, if it can make a decision, whether the representation of the choice is maintained in a high firing attractor state. Such a simple change could determine whether the aggregate activity recorded from such a network resembled the pattern seen in humans or in macaques (Fig. S10). It is quite plausible that simple features of neurons in networks might vary across species and that their activity may be modulated differently depending on whether primary or secondary reinforcers are used⁴⁸.** Either way, the pattern of results is important because it underlines the need for care in interpreting “task-negative” brain areas in human neuroimaging studies⁴⁹. In addition, the results also suggest a similar need for caution when interpreting the fact that activity in other brain areas in human neuroimaging experiments is task positive and increases with task difficulty^{43,50}. **Both types of regions may contain value representations that can guide decision-making.**

In addition supplementary Fig. S10 continues this argument as follows:

Figure S10. Related to Figure 3-5, S6 and S8. Neural network model. (a) Whether activity is positively or negatively related to difficulty may depend on basic features of network⁴⁴. Models such as drift diffusion and biophysical attractor models have shared features and predict which signals should arise in vmPFC and in which order but they do not predict the sign of BOLD changes. If activity in the network is maintained only until decisions are taken then decision difficulty will be associated with increasing BOLD activity but if activity in the network is

maintained even after the decision is taken then decision difficulty will be associated with decreasing BOLD activity.

The figure shows activity in a neural network reflecting the relative evidence in favor of the chosen option as opposed to the unchosen option over time. Colors indicate decisions with increasingly close value differences and hence increasing difficulty. When the decision is difficult (low value difference, blue) the network takes longer to move to an attractor state in which a single population is active (point at which the straight lines peak). If network activity decays once the decision is made (approximately vertical red, green, and blue lines in top part of figure) then the integrated index of activity that is reflected in the sluggish BOLD signal is greater when the decision is difficult; left bottom bars indicate expected BOLD response for the decision which predominantly reflects the time period during which the network moves into an attractor state (difficult decisions, longer integration time, stronger BOLD signal). Such a pattern, where value difference is negatively related to the BOLD response, is seen in macaques in Fig. 3 and 5 and Fig. S6. If, however, the attractor state is maintained even after the decision is taken then, instead of decaying, activity will be as shown by the dashed lines in the top part of the figure. Such a pattern of activity is plausible because biophysical networks of this type have been used to model the maintenance of memory states over short time periods⁴⁷. Now the activity index produced by integration of the sluggish BOLD signal will also reflect the post-decision time. The post-decision period is reached soonest when the decision is easy and choice values are far apart. Now BOLD levels associated with the three decisions are approximated by the length of the shaded bars on the right-hand side of the bottom part of the figure. Easy decisions are now associated with the greatest aggregate activity. Such a pattern is seen in humans in Fig. S6.

Alternatively, the initial, pre-decisional network activity may be higher in macaques compared to humans and hence one pool of neurons may need to be suppressed until an attractor state is reached. Such a model would also predict a negative value difference signal and is also consistent with our observation that the main effect of taking a decision is positive in monkeys but negative in humans (Fig. 3a).

[10] I am not against catchy titles but the one used here seems too generic and does not convey the main message of the paper clearly. Please consider revising the title.

As requested we have changed the title to:

“Less is more: an “inverted” activity pattern in ventromedial prefrontal cortex during value-guided decision-making”

Reviewer #2 (Remarks to the Author):

In this paper Papageorgiou and colleagues present a very interesting set of results, gained via a combination of behavioural testing, lesions and fMRI in macaques. The authors' aim is to dissect the role of vmPFC in guiding value-based choice, and I'm full of praise for the way in which they have tackled this issue by combining different experimental techniques. The paper is timely: very little work has been done on mOFC/vmPFC and economic choice in non-human primates, in particular using fMRI (this is surprising given the vast number of fMRI studies studying the role of vmPFC during value-based choice in humans). The authors' two key results are striking and counter-intuitive when compared with our existing knowledge of human primates. Firstly, they report that the animals systematically prefer the inferior high-value option (HV) to the superior combined option (CV) that includes the HV together with a low valued option (LV). Secondly, they show that the BOLD signal in the macaques' vmPFC codes chosen value vs unchosen value, reversing the usual coding pattern isolated in humans.

We thank the reviewer for their supportive comments. We have attempted to deal with each of the points raised by the reviewer as shown below. In this document the reviewer's comments are shown in bold font, our replies are in italics, and text from the manuscript is shown in a standard font with some new areas of text highlighted in red.

The paper, however, currently struggles to present these results in a coherent fashion. It feels rather like one is reading two papers combined into one. For example, the title and the abstract focus primarily on the behavioural findings (what the authors term the 'less is more effect'), while the vast part of the discussion focuses on the inverse BOLD pattern for value-coding, with little mention of the behavioural findings. This is not just a stylistic problem of how the findings are presented but it runs more deeply in the way analyses are conceived.

Following the reviewer's suggestion we have rewritten the Title, Abstract, and Introduction so that they focus on the finding of neural activity related to subjective value-guided decision-making and the impact of lesions on value-guided decision-making. In the revised manuscript the "less is more" effect is introduced as a way of separating the subjective value of an option from its objective value. Such an argument was made in the original manuscript but it has now been strengthened and additional discussion of the effect has been curtailed.

In addition we have followed the advice of another reviewer, reviewer 1, to focus the Introduction on the fact that no neuroimaging studies have examined the role of vmPFC in the macaque even though a great many have considered the corresponding area in human subjects. Given reviewer 2's comments that "The paper is timely: very little work has been done on mOFC/vmPFC and economic choice in non-human primates, in particular using fMRI (this is surprising given the vast number of fMRI studies studying the role of vmPFC during value-based choice in humans)" we hope that these additional changes will appeal to reviewer 2 too.

The revised title is now:

Less is more: an "inverted" activity pattern in ventromedial prefrontal cortex during value-guided decision-making

The revised abstract now reads:

Ventromedial prefrontal cortex (vmPFC) has been linked to choice evaluation and decision-making in humans but understanding the causal role it plays is complicated by the fact that little is known about the corresponding area of the macaque brain. We recorded activity in macaques using functional magnetic resonance imaging (fMRI) during two very different value-guided decision-making tasks. In both cases vmPFC activity reflected subjective choice values during decision-making just as in humans but the relationship between the blood oxygen level dependent (BOLD) signal and both decision-making and choice value was inverted and opposite to the relationship seen in humans. In order to test whether the vmPFC activity related to choice values was important for decision-making we conducted an additional lesion experiment; lesions that included the same vmPFC region disrupted normal subjective evaluation of choices during decision-making.

The revised Introduction now reads as follows:

Human ventromedial prefrontal cortex (vmPFC) activity covaries with the value of attended objects and potential choices^{11–16}. Moreover, vmPFC activity reflects the key variable that should guide decisions: the difference in value between one choice and another^{17–26} or the value of the default choice^{27,28}. A better understanding of various cognitive processes and their relationship with activity patterns recorded with neuroimaging techniques such as functional magnetic resonance imaging (fMRI) can be gained by using the same fMRI approach in other species, such as macaque monkeys^{29,30}. In macaques, fMRI recording can also be combined with intervention approaches such as lesions to establish the causal importance of the area for a cognitive process. We attempt to do the same here for the case of the vmPFC and value-guided decision-making.

It should be possible to clarify the nature of the contribution vmPFC makes to representation of reward and value and to decision-making by examining the activity of neurons in the homologous area in animal models or by examining the effect of circumscribed lesions. There has, however, been uncertainty about the identity of the human vmPFC region in which activity reflects choice value and its correspondence to brain areas in other primates³¹. In macaque neural activity in an adjacent and partially overlapping region, orbitofrontal cortex (OFC), is more protracted when it is difficult to identify the better of two options because they are close in value³² but few investigations of more medial areas, medial OFC or vmPFC have been conducted. Moreover, surprisingly lesions in the same region do not impair decisions between rewarded and unrewarded stimuli^{33,34}.

In order to understand the nature of vmPFC/OFC activity and its relation to decision-making we carried out a series of experiments in macaques. We based the experimental design and analysis on a series of experiments in humans that have studied a diverse range of decisions but which have been consistent in examining activity related to a key decision variable: the difference in value between the choice taken and the choice rejected during a decision^{17–26}. When this analysis approach is taken, activity is consistently found in an arc-shaped region of human vmPFC corresponding to the region Mackey and Petrides identify as area 14^{35–37}. In human fMRI experiments great care has been taken to show that vmPFC activity is correlated with the subjective value of the choices being considered. Therefore, in experiment 1, we devised a novel behavioral paradigm that allowed separation of the subjective value of choices from the objective amount of reward with which they were associated. In addition, in the same experiment, we show that lesions that include the vmPFC/OFC area disrupt performance of this type of value-guided decision. A variant of this new paradigm is then studied with fMRI in experiment 2. Finally, in experiment 3 we used a different task in which the options' reward values drifted over time and which included task features resembling those of human neuroimaging experiments. We were able to confirm that the findings from experiment 2 could be replicated using a different reward-guided learning task.

For example, let's assume that the authors want to focus on understanding the computational underpinnings of the 'irrational' behaviour that they isolated. There are a number of questions that one would like to see addressed to understand this phenomenon more deeply. For example, in the HV vs CV, does the LV option act as a decoy when presented in combination with the HV, thereby lowering the value of HV?

Note that there is a vast literature on this phenomenon in the field of JDM that the authors do not mention. Is the order in which the options are presented in the CV case important?

We thank the reviewer for raising this suggestion. In the revised manuscript we have now discussed this possibility. In brief, however, note the effect does not depend on the order in which the CV option's component parts are presented. In experiment 1, the "less is more" pattern of decision-making occurs at a time point when the monkeys can only see the conditioned stimulus, the CS+, but not the unconditioned stimuli, the fruit and vegetable. If the monkey chooses the CV CS+ then the monkey is given both component parts of the outcome simultaneously.

In experiment 2 the effect does not depend on the order in which the CV options are presented. Again the "less is more" pattern of decision-making occurs at a time point when the monkeys can only see the CS+ and not the juice drop outcomes. If the monkey chooses the CV CS+ then the two types of juice that comprise the outcome are delivered sequentially but in an order that is counterbalanced across trials to avoid any effect of the delivery order of the reward components. On half of occasions the high value component of the CV outcome is presented first. On half of occasions the low component of the CV outcome is presented first. In summary the order in which the components that comprise the CV outcome cannot explain the less is more effect.

In order to deal with this important point raised by the reviewer we have revised the manuscript so that it now reads:

"Control macaques' choices on HV-CV decisions appear irrational and suggest subjective CV value estimates are not optimal. **One possibility is that the monkeys' estimation of the CV option is biased away from the sum of the component parts towards their mean.** Similarly, humans sometimes average values of groups of items instead of summing them ³⁹; collectors paid more for sets of high value baseball cards than for identical sets of high value cards with additional low value cards. **In another experiment humans valued a set of dinnerware less even if it were larger if it also contained additional broken items⁴⁰.** A related pattern of behavior has been reported, albeit in one macaque, previously studied in a task sharing features with those used in experiments 1 and 2 ⁴¹. By contrast, animals with vmPFC/OFC lesions were less apparently irrational. While this may appear to conflict with a vmPFC role in decision-making ⁴² it is important to note that the lesion-associated difference in behavior occurs because monkeys become increasingly indifferent between the HV and CV options while control animals have clear but counterintuitive preferences.

One possibility is that the monkeys preferred the CV option to the HV option because of some aspect of the way in which the outcomes were ordered when they were delivered. This seems unlikely because the two component parts of the CV outcome were delivered simultaneously in experiment 1 while in experiment 2, although presented sequentially, the CV outcome order was counterbalanced across trials. Another possibility is that the "less is more"

effect is due to the food or juice becoming distasteful when two types are available in the same outcome. However, the food/juices of the CV option were offered sequentially and they were not mixed. Moreover the sequential consumption of different types of rewards is not very likely to be problematic; foraging animals consume various types of foods in sequence when they are hungry. In experiment 1 the outcomes were presented separately but simultaneously and the animals decided when, how, and in what order to put the items into their mouths. Moreover, if consuming the foods/drinks comprising the CV option sequentially were distasteful then the animals should not have chosen the CV option when this was contrasted with the LV option.

The choice pattern of control animals may appear more rational if one remembers that decisions in the real world are often made between several multi-component options in contexts where each component outcome is only probabilistically rather than deterministically linked to the animal's choice. In such a scenario evaluating options in terms of their mean value would rarely be detrimental and possibly even efficient. Moreover if decision-making is seen within the broader context of the foraging decisions macaques evolved to take⁴³; then the added value of CV options may be outweighed by the handling costs of the LV component and the cost incurred by failing to move on to other opportunities."

From a more neuroscientific angle, is this effect driven by normalisation? (see for example the work of Louie, Glimcher or Soltani).

We thank the reviewer for these suggestions. Arguably a conventional model of value normalization cannot explain the pattern of decisions between HV and CV options that we observed. The conventional model of value normalization involves a space-based code in which an option that is presented visually (or perhaps via other sensory modalities) would inhibit the neuronal representation of all other options that are presented simultaneously at other parts of the visual field. As such, additional rewards (such as the LV component of the CV option) could only reduce the discriminability between options (such as the HV option and the HV component of the CV option) but not reverse choices in the manner we observed. However, one might hypothesize that normalization could happen in a different way: for multi-attribute choices even though individual attributes within an option were not presented using separate stimuli they might still normalize one another. If such a form of normalization exists then our CV option would have a smaller subjective value than might otherwise have been expected because the value of the HV component within the CV option would be normalized by the presence of the adjacent LV component.

We are not quite sure how to think about the less is more effect in terms of a decoy effect. Very often decoy effect involve decision-making experiments in which three options with multiple attributes are presented simultaneously. If one took an analogous approach to the multiattribute CV option that we have described in relation to normalization effects then we might consider the possibility that a decoy effect might operate between the CV option's component parts. Also, we might consider that the absent option might take the role as a decoy in our experiments that involved two options at a time (e.g. the LV option is a decoy for decisions between CV and HV). There are three kinds of decoy effects that are widely discussed – similarity, compromise and dominance. The similarity and compromise effects describe how preference is "nudged" by a decoy when all three options are equally preferable and these effects would be irrelevant to our case. The dominance effect, however, would make an opposite prediction to our behavioral results. In decisions between HV and CV, one would predict that the value of the HV component within the CV option would be enhanced by the comparison with the LV option. According to this logic the CV option would become even more preferred over the HV option, which is opposite to our empirical data.

In summary, it is possible that the less is more effect might be understood in relation to a normalization effect although an explanation using this approach is not straightforward. We

have therefore mentioned it in the Discussion in the revised manuscript and cited papers by the authors that the reviewer suggested as follows:

An alternative way of thinking about the biasing of the CV option's value towards the mean of its components might make reference to value normalization⁵²⁻⁵⁶. According to such a view the biasing of the CV option's value towards the mean value of its component parts might occur if the monkey's attentional focus is on the HV component within the CV option but if the HV's value is normalized by the presence of the adjacent LV component. An HV option presented in isolation would not be normalized in the same manner.

Moreover, it is unclear via which mechanism the lesion reduces (but does not abolish) this effect. It is also puzzling why the lesion seems to have little effect on the ability to discriminate the most rewarding option. Is this due to the fact that in this type of task the animals are heavily trained on the association between fractal and reward, making this task more dependent on a striatal circuit and less on OFC (that is usually involved in constructing on-the-fly values)?

We think that it is not puzzling that the lesion does not abolish the ability of the animals to discriminate the most rewarding option when it is offered with a low value option because similar lesions in non-human primates do not impair simple reward-guided visual discrimination. However, we think that the reason why this is the case is probably along the lines suggested by the reviewer: other brain areas such as the striatum or perhaps the perirhinal cortex may be important for linking stimuli with rewards. We think that other brain areas such as central parts of the orbitofrontal cortex may be involved in "on-the-fly" construction of a value given changes in satiety⁵⁷. We think, however, that more medial parts of the OFC and adjacent vmPFC may be involved in the construction of a value that is constantly being updated (experiment 3) or which consists of different elements (experiments 1 and 2). In the revised manuscript we have written:

VmPFC/OFC lesion effects

Lesions of vmPFC/OFC in macaques have comparatively little effect on reward-guided visual discrimination in many circumstances^{33,34}. One interpretation of such a pattern of results is that vmPFC/OFC does not play a critical role in value-guided choice. An alternative interpretation, however, is that performance of many simple reward-guided visual discrimination tasks may be mediated by representations of stimulus-reward association in other brain regions such as perirhinal cortex⁵⁸ or the striatum^{59,60}. It is possible, however, that a task that separates subjective values from objective reward amounts, such as the one that we have devised, will be affected by vmPFC/OFC lesions. (p4)

The lesion results (Fig. 1) suggest vmPFC activity may be essential for decisions guided by subjective value estimates. The lesions we studied changed the way that animals chose between HV and CV options but did not disrupt the ability to distinguish the HV from the LV option. This is consistent with observations that lesions in this region have little impact on the ability to learn and use simple stimulus reward associations and to learn which option is rewarded⁶¹. Instead such simple stimulus-reward associations may depend on other brain regions such as the striatum^{59,60} and rhinal cortex⁵⁸. The lesions may have compromised the connections of vmPFC³⁴ and, in addition to areas 11m and 14, the lesions included adjacent areas 11 and 13 in the central orbital region between the medial and lateral orbital sulci. The fMRI results, however, enable identification of regions within the lesion zone that are closely related to the task; they emphasize activity in vmPFC areas 14 and 11m. Nevertheless, it is

important to note that the statistical thresholding used in establishing fMRI activations as significant is conservative. For example, at lower thresholds activity in our experiments was typically more bilateral although it did not extend beyond the lesion zone. fMRI is particularly sensitive to changes in aggregate activity and it is likely that more widespread activity is associated with the reward-guided decision process throughout adjacent OFC areas^{32,62–64}. Central OFC areas 11 (as opposed to 11m) and 13 may be most important when it is necessary to estimate values “on the fly” from “knowledge of the causal structure of the environment”^{33,65} while more medial vmPFC may be important for decisions guided by value estimates that are constantly updated from experience (experiment 3) or which are constructed from different component elements (experiments 1 and 2). (p14)

Can the devaluation experiment shed some light on this? In other words, if this is the central topic of the paper (as the title suggests), I would suggest that the authors look into this more deeply and discuss in more detail the theoretical implications of their findings (especially given the fact that the last author has authored many insightful papers on this topic).

We thank the reviewer for this helpful suggestion that greater discussion of the devaluation procedure might clarify whether the reward values are overlearned. In the revised manuscript we note that the value of the LV stimuli was subject to revision when the LV outcome was devalued by pre-feeding and so the LV stimuli may not be overlearned. However, we note that our aim in conducting the devaluation procedure was different than in some other studies. Our aim was simply to lower the value of the LV stimuli to see if we could make the effect that we had observed larger. For example we did not test whether LV devaluation occurred even when LV choices were tested in extinction as is done in experiments in which the goal is to determine if the LV values are “goal-based” values that inferred “on the fly”. In the supplementary information we note:

Reward Devaluation – Experiment 1

After completing the last three days of the main task, all monkeys were given revision sessions (S11) that included only the HV option, one of the two LV options, and the corresponding CV option. Once animals reached the learning accuracy criterion they continued with the Devaluation phase (S12).

Devaluation procedures are often used in the context of investigations of goal-based decision-making and the role of the OFC^{23,34,65–68}. In such procedures animals are allowed to feed to satiety on one food type so that its value to the animal decreases. In other variants on the procedure, the food item is devalued by pairing with nausea. Critically, when devaluation is used to assess goal-based decision-making, it is imperative that animals make choices between CS+s associated with each food type without experiencing the food itself subsequently; this is necessary to ensure that the animals are making choices on the basis of internal representations of the expected outcomes that have been revalued in the absence of direct experience of the particular choice and the outcome in the new sated state. In order to ensure that this is the case studies with rodents often examine decision-making during extinction when rewards are not actually delivered to the animals^{65,67,68}. Studies of goal-based decision-making conducted in macaques avoid giving macaques repeated experience of a given CS+ choice in the context of the food type by training the monkeys on multiple pairs of stimuli and ensuring that any given CS+ is only experienced once during the critical decision-making test when CS+s associated with each of the two foods are paired against one another^{23,34,66,69}.

By contrast, here the aim of the devaluation procedure was quite distinct. It was not intended to investigate goal-based decision-making on the basis of values inferred “on the fly”.

Instead the aim was quite simply to decrease the value of the vegetable option. We were interested in investigating the possibility that the CV option, which was partly comprised of the vegetable option, would become even less valuable than the HV option than had previously been the case. The devaluation session was conducted ~24h after the previous feeding opportunity (so that more than one food type was not inadvertently devalued). A food box (22 x 10 x 14.5cm) was placed in the monkeys' home cage filled with 250g of the vegetable option that was to be devalued. The monkey was free to consume the food for 25 minutes without being directly observed. The experimenter then entered the room and if most of the food was consumed, an additional 150g of the same food was added. After five minutes the experimenter started observing the animal through the monkey's housing room window until the monkey refrained from consuming any food for five minutes. The food box was then removed from the home cage and the remaining food was weighed at the end of the session. For all monkeys, 35 minutes were sufficient to complete this procedure. The monkey was then taken from its home cage and moved into the testing room in a transport box. The main phase of the testing started within five minutes from the completion of the selective satiation procedure.

During this phase, a task similar to the one described in *Main task* was given. Table S1 provides a schematic representation of the different stages of the experiment.

On the other hand, if the goal is to understand whether and why the value-coding is reversed compared to humans (and possibly discuss this in relation to the resting state signal which is not reversed), why not focus on the simpler cases in which chosen and unchosen are constructed using LV vs HV, and in which the behaviour is straightforward and in line with the prediction of classical decision theory?

We understand from the reviewer's comments that there is confusion about the main focus of our manuscript. We have therefore revised the manuscript extensively. In doing so we have followed the reviewer's suggestions. We hope, however, that our manuscript may have been made clearer by following the suggestion made by reviewer 1 for revising the Introduction. In brief, reviewer 1 suggested that we explain the rationale for the experiments in a simpler manner given that there are no other fMRI studies of reward-guided decision-making in macaques. We hope that this will accord with the general spirit of Reviewer 2's comments regarding the need for a more straightforward manuscript. In summary in the revised manuscript we note simply that there is a large body of neuroimaging studies investigating reward-guided decision-making in humans that emphasizes activity that reflects the subjective value of choices. In the revised manuscript we explain that the "less is more" procedure in experiment 2 means that we are able to identify activity that is related to the subjective value of an outcome rather than its objective amount. We note that by using gradually drifting reward rates in experiment 3 we are using an approach that is similar to that used in many human experiments.

However, we have also complied with the suggestion of the reviewer and replicated one of our neural key results based on trials without the CV. This assures that our results also hold for the simpler case where chosen and unchosen values are constructed using only LV and CV. This is included in a supplementary figure as follows:

Figure S5. Related to Figure 3. Control analyses for “inverse” value activity pattern. (a) Value-related activity at the decision-related ROI (GLM-1) and **(b)** in choices in which the CV option is not available (GLM-2), in experiment 2. ... **(b)** The regression coefficients relating the BOLD signal to the difference between chosen and unchosen options at the time of choice in experiment 2 is plotted in the same manner as in figure 3e but now trials in which the CV option was offered have not been included in the analysis. This means that the chosen-unchosen regression coefficients are based on data from trials on which only the single component HV, LV, and unrewarded options were offered to animals.

I am worried that during the HVvsCV trials other processes might be going on (see point above). For example, let us assume that the effect reported in Figure 3 (e-f) is mostly driven by the CV_vs_HV trials: one might claim that vmPFC is coding the ‘underpinning’ unbiased value that should be chosen (i.e. CV > HV) in the same fashion to the one shown in human studies, even if then (for reasons that are unclear - see previous paragraph) the subjects chose the ‘low value’ CV option. I see that the authors have already made some comparison between HV and LV but only on the overall signal (Figure 4); however, I could not find a similar analysis for the chosen vs unchosen. I suspect that the last experiment which uses probabilistic reward might help to clarify this issue, but I would like to see this unpacked more in the manuscript.

First, a simple and direct way to address reviewer 2’s point with an additional analysis is to make a new version of Fig. 3e and 3f that uses the subset of trials on which the CV option was not present as either the chosen or unchosen option (see above). In the revised manuscript this is what we have done. We now include an additional figure, Fig. S5b that illustrates such an analysis.

Second, we contend that Fig. 4 may already address the reviewer’s concerns because it shows that activity related to the HV option when it is likely to be chosen against either LV or unrewarded alternatives (green and red lines in panel a) and the LV option when it is chosen against the unrewarded option (panel c). It should be clear that vmPFC reflects chosen and unchosen values even when the CV option is not one of the options available to choose on a given trial.

Third, we note that experiment 3 does not contain compound options but the vmPFC still tracks its activity.

In summary the revised manuscript now reads:

One possibility is that the vmPFC activity pattern reflects some unusual feature of the CV option in which the subjective value was dissociated from the objective value. In order to test whether this is the case we took three additional measures. Most importantly, we carried out an additional experiment (experiment 3) that we describe below that eschewed the use of a CV option. In addition, however, we carried out additional analyses identical to those shown in Fig. 3e and f but we only included data from trials on which the CV option had not been available. Once again, we saw a very similar pattern of activity related to the value of the choice taken and the value of the choice rejected (Fig. S5b).

Another way to check that the interpretation of the vmPFC activity identified by the various parametric GLM analyses described above is not unduly affected by the presence of the CV option is to examine activity related to the presence of each of the choice options (HV, CV, LV, and Unrew) in a complementary analysis (Experimental Procedures; Table S3: GLM-1, contrasts based on all cue-onset regressors and sorted by choice taken; Fig. 4). Because the analyses shown in Fig. 3d- f already suggest that the manner in which the presence of the HV, CV, LV, or Unrew option affects vmPFC activity depends on whether or not it is chosen we cannot look simply at trials containing the HV, CV, LV, or Unrew options; in addition we must consider the context in which it was presented (what was the other option presented). This complementary analysis had the advantage that it did not depend on precisely how values were assigned to each choice because we simply looked at activity on each of the main decision types. To perform the analysis we took the peak activity from each trial within a 5sec period between 1.5 s and 6.5 s after stimulus presentation (Fig. 4d). It revealed that vmPFC activity reflected the subjective values, rather than the objective reward amounts associated with the stimuli. **This was true regardless of whether the options were simple options such as HV and LV or compound options such as CV.** Moreover, vmPFC increased as the subjective value of the chosen option decreased (compare green and red lines in panels moving left to right; Fig. 4a: HV choices; Fig. 4b: CV choices) and increased as the subjective value of the unchosen option increased (compare red, green, and blue lines; Fig. 4a: HV choices; 4b: CV choices); in a two (chosen value: CV, HV) by two (unchosen value: Unrewarded, LV) factorial ANOVA there was an effect of chosen value ($F_{1,3}=15.236$, $p=0.03$; **after square-root transformation: $F_{1,3}=36.482$, $p=0.009$**) and unchosen value ($F_{1,3}=10.375$, $p=0.049$; **after square-root transformation: $F_{1,3}=21.740$, $p=0.019$**). Such a pattern suggests greater aggregate vmPFC activity when identifying the better option was difficult because it had a low value or because the alternative option had a high value and inspection of the CV-related activation patterns confirms that it is the subjective value of choices, rather than objective reward amount that is correlated with vmPFC activity.

In addition, Fig. S5b now shows the chosen-unchosen value effect in experiment 2 calculated in the absence of CV option trials as follows:

Figure S5. Related to Figure 3. Control analyses for “inverse” value activity pattern. (a) Value-related activity at the decision-related ROI (GLM-1) and **(b)** in choices in which the CV option is not available (GLM-2), in experiment 2. **(a)** The difference in value between the chosen and unchosen option values (shown in green color) had a significant impact on vmPFC activity (after square root transformation: $t_3=-3.4307$, $p=0.041$) in an ROI around the peak of the main effect of decision-making (Fig. 3a). Such a pattern of activity suggests the main effect of taking a decision (activity that changes whenever a decision is taken regardless of the values of the options considered; shown in Fig. 3a, b) is found in vmPFC in the same regions as activity related to the key variable – the difference in value between the choice taken and the choice rejected – that should drive each decision (shown also in Fig. 3d-f). **(b)** The regression coefficients relating the BOLD signal to the difference between chosen and unchosen options at the time of choice in experiment 2 is plotted in the same manner as in figure 3e but now trials in which the CV option was offered have not been included in the analysis. This means that the chosen-unchosen regression coefficients are based on data from trials on which only the single component HV, LV, and unrewarded options were offered to animals.

In conclusion, I believe that the authors potentially have a strong and in part surprising set of findings in their hands. I believe that these findings might spark great interest in the field, but for this very reason, I strongly recommend them to revise the manuscript to make their message less ambiguous.

We thank the reviewer for noting that the results are potentially interesting. By following the suggestions made by reviewer 2 we hope that we have made the manuscript clearer.

Reviewer #3 (Remarks to the Author):

This manuscript describes results from behavioral, lesion and fMRI experiments on macaque monkeys performing a reward-based decision making task. Subjects made choices between cues predicting a high value fruit outcome (HV), a lower value vegetable outcome (LV), and a combination of the fruit and the vegetable (CV=HV+LV). As expected, monkeys preferred HV over LV and CV over LV cues. Strikingly, however, monkeys preferred HV over CV cues, even though CV cues predicted more reward in absolute terms. Lesions to the ventromedial prefrontal cortex (vmPFC) and orbitofrontal cortex partially changed this pattern of behavior, such that lesioned monkeys still prefer HV over CV cues, but to a lesser degree than control animals. Imaging data from a similar task involving fruit and vegetable juice indicated that BOLD signals in the vmPFC encode the difference in value between the chosen and the unchosen cue. However, in contrast to a large number of studies on human vmPFC, BOLD responses were negatively (and not positively) related to this value difference.

This is an extremely interesting and important study. The results will be of broad interest and are likely to stimulate much follow-up comparative research. I have several comments that should be addressed.

We thank the reviewer for their supportive comments. We have attempted to deal with each of the points raised by the reviewer as shown below. In this document, the reviewer's comments are shown in bold font, our replies are in italics, and text from the manuscript is shown in a standard font with some new areas of text highlighted in red.

1. The behavior is remarkable. The authors convincingly show that monkeys prefer cues predicting single high-value rewards over cues predicting the same high-value reward in combination with an additional low-value outcome. There are two potential explanations for this behavior. Either (as the authors suggest) monkeys average the predicted value of the two cues and base their choice on this average value. Alternatively, the value of the two outcomes, when received together, is lower than the sum of the two rewards when received in isolation (somehow, combining a vegetable and fruit ruins the dish). In this case, monkeys would not average two independent value predictions but instead use the value of the combined reward to guide their choice. This is a subtle but theoretically important distinction. To dissociate between these two possibilities the authors could present choice data from situations in which monkeys choose directly between the fruit vs the fruit & vegetable combination (without cues). I am not sure whether this is what the authors refer to in the sentence on page 4: "This was the case even though macaques took both components parts of the CV outcome." Data on this would be essential.

The reviewer is making an important distinction in relation to a matter in which we are also interested. It is, however, unlikely that the less is more effect can be attributed to the animals being "reluctant to mix their food/drinks in one go" in any simple way. The food/juices of the CV option are offered sequentially and they are not mixed. Moreover the sequential consumption of different types of rewards is not very likely to be problematic; foraging animals consume various types of foods in sequence when they are hungry.. We have clarified these points in the revised manuscript. In addition, however, in the revised manuscript we do make clear that the evaluation of the CV option may appear more rational if decision-making is seen within the broader context of the foraging decisions animals evolved to take³⁸. The HV and LV

components that comprise the CV each have a value but they also have a handling cost which may increase when the components are offered together as a single outcome. It is possible, perhaps, to think of such an explanation as a slight variant on the reviewer's suggestion that "somehow, combining a vegetable and fruit ruins the dish").

The reviewer also asks about the degree to which the effect might be restricted to choices made between CSs associated with foods or whether it is present when choices are made between unconditioned stimuli such as the foods themselves. That is also a very interesting suggestion and we agree that it would be worth investigating in a future experiment. However, we are not in a position to run further experiments with these animals. Moreover even if we were, it would be difficult to perform such an experiment in the MRI scanner too. Finally, we note that the external examiner of the first author's doctoral thesis was also interested in the same question and has collected some preliminary data on this issue. We therefore hope that this will be published in due course.

In an attempt to deal with the reviewer's comments we have revised the manuscript as follows:

"Control macaques' choices on HV-CV decisions appear irrational and suggest subjective CV value estimates are not optimal. **One possibility is that the monkeys' estimation of the CV option is biased away from the sum of the component parts towards their mean.** Similarly, humans sometimes average values of groups of items instead of summing them ³⁹; collectors paid more for sets of high value baseball cards than for identical sets of high value cards with additional low value cards. **In another experiment humans valued a set of dinnerware less even if it were larger if it also contained additional broken items⁴⁰.** A related pattern of behavior has been reported, albeit in one macaque, previously studied in a task sharing features with those used in experiments 1 and 2 ⁴¹. By contrast, animals with vmPFC/OFC lesions were less apparently irrational. While this may appear to conflict with a vmPFC role in decision-making ⁴² it is important to note that the lesion-associated difference in behavior occurs because monkeys become increasingly indifferent between the HV and CV options while control animals have clear but counterintuitive preferences.

One possibility is that the monkeys preferred the CV option to the HV option because of some aspect of the way in which the outcomes were ordered when they were delivered. This seems unlikely because the two component parts of the CV outcome were delivered simultaneously in experiment 1 while in experiment 2, although presented sequentially, the CV outcome order was counterbalanced across trials. Another possibility is that the "less is more" effect is due to the food or juice becoming distasteful when two types are available in the same outcome. However, the food/juices of the CV option were offered sequentially and they were not mixed. Moreover the sequential consumption of different types of rewards is not very likely to be problematic; foraging animals consume various types of foods in sequence when they are hungry. In experiment 1 the outcomes were presented separately but simultaneously and the animals decided when, how, and in what order to put the items into their mouths. Moreover, if consuming the foods/drinks comprising the CV option sequentially were distasteful then the animals should not have chosen the CV option when this was contrasted with the LV option.

The choice pattern of control animals may appear more rational if one remembers that decisions in the real world are often made between several multi-component options in contexts where each component outcome is only probabilistically rather than deterministically linked to the animal's choice. In such a scenario evaluating options in terms of their mean value would rarely be detrimental and possibly even efficient. Moreover if decision-making is seen within the broader context of the foraging decisions macaques evolved to take ⁴³; then the added value of CV options may be outweighed by the handling costs of the LV component and the cost incurred by failing to move on to other opportunities."

2. Even more astonishing are the findings from the imaging data. The authors, again convincingly, show that BOLD responses in the macaque vmPFC are negatively related to the value of the chosen option and positively related to the value of the unchosen option. This is in direct opposition to a large number of findings in humans. The authors do not offer a particularly compelling explanation for this discrepancy other than citing that both signatures are generally compatible with neural processes related to value computation. Even though I agree with the assessment that both signatures are in line with value computation, how is it possible that two species that are remarkably similar in many aspects of brain function somehow show the opposite BOLD effects in this task? I think it would be premature to just accept this as an effect of “species differences”. I believe the authors should thoroughly explore and discuss alternative explanations for this discrepancy such as differences in task, training, data analysis, etc.

In the revised manuscript we have attempted to deal with this point more thoroughly. We note that several studies have shown that activity in areas such as vmPFC can be captured by a variety of computational models of decision-making such as biophysical cortical attractor models⁴⁶. Such models predict which variables should affect activity and when such influences should arise but they do not make strong predictions about the sign with which BOLD changes should occur; the sign may reflect features of the network that are not integral to the decision process itself but which are related to whether choice representations are maintained only until decisions are taken or if they are maintained subsequently. We explain that attractor models⁴⁶ contain populations of neurons and each represents one possible choice. During the decision process, the network moves to an attractor state in which just a single population reaches a high-firing state. In such a model the high firing attractor state may or may not decay quickly after a decision is reached and this simple difference may be sufficient to flip the sign of a value difference signal recorded with fMRI (see Fig. S7 for further discussion). We explain that simple features of neurons in recurrent networks related to their resting activity levels or to the degree to which activity is maintained in a high firing attractor state could therefore produce the activity patterns seen in monkeys or humans. For example Wong and Wang⁴⁷ have described how a change a single network parameter, the levels of recurrent excitation, can determine whether or not the network can make a decision and, if it can make a decision, whether the representation of the choice is maintained in a high firing attractor state. Such a simple change could determine whether the aggregate activity recorded from such a network resembled the pattern seen in humans or in macaques. It is quite plausible that simple features of neurons in networks might vary across species and that their activity may be modulated differently depending on whether primary or secondary reinforcers are used⁴⁸.

In other words, by changing a single parameter of a neural network such as the amount of recurrent excitation it is possible to change the length of time for which a choice is represented – is it represented just during the decision period as the network moves to a high firing attractor state or is it maintained in a high firing attractor state subsequently. Such a change is sufficient to alter the sign of the regression weight of value of the BOLD signal. The revised manuscript argues as follows:

Although the sign of activity changes is given great weight when interpreting function in human neuroimaging, not just in vmPFC but in other areas such as dorsal anterior cingulate cortex⁴³, whether activity is positively or negatively related to difficulty may simply depend on basic features of the networks mediating decision-making⁴³. Several studies^{19,20,22,26,44,45} have shown that activity in frontal lobe areas such as vmPFC can be captured by a variety of computational models of decision-making, which share several features, such as drift diffusion

and biophysical cortical attractor models⁴⁶. Such models predict which variables should affect activity and when such influences should arise but they do not make strong predictions about the sign with which BOLD changes should occur; the sign may reflect features of the network that are not integral to the decision process itself but which are related to whether choice representations are maintained only until decisions are taken or if they are maintained subsequently.

For example attractor models⁴⁶ contain populations of neurons and each represents one possible choice. During the decision process, the network moves to an attractor state in which just a single population reaches a high-firing state. In such a model the high firing attractor state may or may not decay quickly after a decision is reached and this simple difference may be sufficient to flip the sign of a value difference signal recorded with fMRI (see Fig. S10 for further discussion). **Simple features of neurons in recurrent networks related to their resting activity levels or to the degree to which activity is maintained in a high firing attractor state could therefore produce the activity patterns seen in monkeys or humans. For example Wong and Wang⁴⁷ have described how a changes in a single network parameter, the level of recurrent excitation, can determine whether or not the network can make a decision and, if it can make a decision, whether the representation of the choice is maintained in a high firing attractor state. Such a simple change could determine whether the aggregate activity recorded from such a network resembled the pattern seen in humans or in macaques (Fig. S10). It is quite plausible that simple features of neurons in networks might vary across species and that their activity may be modulated differently depending on whether primary or secondary reinforcers are used⁴⁸.** Either way, the pattern of results is important because it underlines the need for care in interpreting “task-negative” brain areas in human neuroimaging studies⁴⁹. In addition, the results also suggest a similar need for caution when interpreting the fact that activity in other brain areas in human neuroimaging experiments is task positive and increases with task difficulty^{43,50}. **Both types of regions may contain value representations that can guide decision-making.**

In addition supplementary Fig. S10 continues this argument as follows:

Figure S10. Related to Figure 3-5, S6 and S8. Neural network model. (a) Whether activity is positively or negatively related to difficulty may depend on basic features of network⁴⁴. Models such as drift diffusion and biophysical attractor models have shared features and predict which signals should arise in vmPFC and in which order but they do not predict the sign of BOLD changes. If activity in the network is maintained only until decisions are taken then decision difficulty will be associated with increasing BOLD activity but if activity in the network is

maintained even after the decision is taken then decision difficulty will be associated with decreasing BOLD activity.

The figure shows activity in a neural network reflecting the relative evidence in favor of the chosen option as opposed to the unchosen option over time. Colors indicate decisions with increasingly close value differences and hence increasing difficulty. When the decision is difficult (low value difference, blue) the network takes longer to move to an attractor state in which a single population is active (point at which the straight lines peak). If network activity decays once the decision is made (approximately vertical red, green, and blue lines in top part of figure) then the integrated index of activity that is reflected in the sluggish BOLD signal is greater when the decision is difficult; left bottom bars indicate expected BOLD response for the decision which predominantly reflects the time-period during which the network moves into an attractor state (difficult decisions, longer integration time, stronger BOLD signal). Such a pattern where value difference is negatively related to the BOLD response is seen in macaques in Fig. 3 and 5 and Fig. S6. If, however, the attractor state is maintained even after the decision is taken then, instead of decaying, activity will be as shown by the dashed lines in the top part of the figure. Such a pattern of activity is plausible because biophysical networks of this type have been used to model the maintenance of memory states over short time periods⁴⁷. Now the activity index produced by integration of the sluggish BOLD signal will also reflect the post-decision time. The post-decision period is reached soonest when the decision is easy and choice values are far apart. Now BOLD levels associated with the three decisions are approximated by the length of the shaded bars on the right-hand side of the bottom part of the figure. Easy decisions are now associated with the greatest aggregate activity. Such a pattern is seen in humans in Fig. S6.

Alternatively, the initial, pre-decisional network activity may be higher in macaques compared to humans and hence one pool of neurons may need to be suppressed until an attractor state is reached. Such a model would also predict a negative value difference signal and is also consistent with our observation that the main effect of taking a decision is positive in monkeys but negative in humans (Fig. 3a).

3. The current version of the manuscript touches several vmPFC-related concepts including value coding, choice difficulty, default mode network, and value computation in neural networks. However, throughout, these concepts are sporadically picked up and then dropped again without following through with a line of argumentation. Also, it is unclear how the main finding (value encoding is inverted in monkeys) is related to any or all of these concepts (except value coding). Some conceptual streamlining would improve the manuscript.

From some of the other reviewer comments we feel that revising the Abstract and Introduction might be particularly helpful for clarifying the manuscript. We have therefore followed the suggestions made by reviewer 1 for streamlining the Introduction. The revised Abstract now reads as follows:

Ventromedial prefrontal cortex (vmPFC) has been linked to choice evaluation and decision-making in humans but understanding the role it plays is complicated **by the fact that little is known about the corresponding area of the macaque brain**. We recorded activity in macaques using functional magnetic resonance imaging (fMRI) during two very different value-guided decision-making tasks. **In both cases vmPFC activity reflected subjective choice values during decision-making just as in humans but the relationship between the blood oxygen level dependent (BOLD) signal and both decision-making and choice value was inverted and opposite to the relationship seen in humans. In order to test whether the vmPFC activity related**

to choice values was important for decision-making we conducted an additional lesion experiment; lesions that included the same vmPFC region disrupted normal subjective evaluation of choices during decision-making.

The revised Introduction now reads as follows:

Human ventromedial prefrontal cortex (vmPFC) activity covaries with the value of attended objects and potential choices^{11–16}. Moreover, vmPFC activity reflects the key variable that should guide decisions: the difference in value between one choice and another^{17–26} or the value of the default choice^{27,28}. A better understanding of various cognitive processes and their relationship with activity patterns recorded with neuroimaging techniques such as functional magnetic resonance imaging (fMRI) can be gained by using the same fMRI approach in other species, such as macaque monkeys^{29,30}. In macaques, fMRI recording can also be combined with intervention approaches such as lesions to establish the causal importance of the area for a cognitive process. We attempt to do the same here for the case of the vmPFC and value-guided decision-making.

It should be possible to clarify the nature of the contribution vmPFC makes to representation of reward and value and to decision-making by examining the activity of neurons in the homologous area in animal models or by examining the effect of circumscribed lesions. There has, however, been uncertainty about the identity of the human vmPFC region in which activity reflects choice value and its correspondence to brain areas in other primates³¹. In macaque neural activity in an adjacent and partially overlapping region, orbitofrontal cortex (OFC), is more protracted when it is difficult to identify the better of two options because they are close in value³² but few investigations of more medial areas, medial OFC or vmPFC have been conducted. Moreover, surprisingly lesions in the same region do not impair decisions between rewarded and unrewarded stimuli^{33,34}.

In order to understand the nature of vmPFC/OFC activity and its relation to decision-making we carried out a series of experiments in macaques. We based the experimental design and analysis on a series of experiments in humans that have studied a diverse range of decisions but which have been consistent in examining activity related to a key decision variable: the difference in value between the choice taken and the choice rejected during a decision^{17–26}. When this analysis approach is taken, activity is consistently found in an arc-shaped region of human vmPFC corresponding to the region Mackey and Petrides identify as area 14^{35–37}. In human fMRI experiments great care has been taken to show that vmPFC activity is correlated with the subjective value of the choices being considered. Therefore, in experiment 1, we devised a novel behavioral paradigm that allowed separation of the subjective value of choices from the objective amount of reward with which they were associated. In addition, in the same experiment, we show that lesions that include the vmPFC/OFC area disrupt performance of this type of value-guided decision. A variant of this new paradigm is then studied with fMRI in experiment 2. Finally, in experiment 3 we used a different task in which the options' reward values drifted over time and which included task features resembling those of human neuroimaging experiments. We were able to confirm that the findings from experiment 2 could be replicated using a different reward-guided learning task.

Experiment 1 is now introduced by explaining that it introduces a paradigm allowing us to separate subjective from objective values as follows:

In the first experiment we present a new paradigm that allows separation of the subjective value of choices from the objective amount of reward with which they are associated. Four control macaques

4. The authors present a number of different analyses of the imaging data and it is at times difficult to follow what a particular analysis is meant to show or add. I wonder whether there are more direct ways to portray the data. For instance, would it be possible to plot parameter estimates from a GLM with HRF-convolved regressors for the different conditions shown in Figure 3 Ci (HV, CV, LV, Unrew)?

First to address the reviewer's concerns about the clarity of the explanation of the various value signals that we find in vmPFC we have tried to explain their meaning and implication of the results more clearly in the revised manuscript. We have tried to provide brief introductions to various result sections and summaries of the various findings that we have made. The Result section of experiment 2 is the most complex and it has now been revised so that it reads:

Increased vmPFC activity at the time of choice

In order to investigate whether general changes in vmPFC BOLD activity were similar in monkeys compared to humans, we focused our initial fMRI analysis at decision-related events. During these events a large cluster with increased activity was found within the region investigated with lesions (Experimental Procedures; Table S3: GLM-2, contrast 1); it extended from the medial orbital sulcus across the gyrus rectus (possibly areas 14r and 11m³⁶; Fig. 3a; cluster corrected $z > 2.3$; $p < 0.05$). This region has sometimes been referred to as medial OFC (mOFC)⁷⁰ but for ease of comparison with humans³⁵⁻³⁷, we refer to it as “vmPFC”. **Just as in human subjects there was a clear effect of decision-making on the BOLD signal. However, while decision-making is accompanied by a decrement in the vmPFC BOLD signal in humans we found that it was linked to a BOLD increment in macaque vmPFC (Fig. 3a, b).** No negative activation cluster was found in vmPFC or adjacent brain areas (Fig. 3g). In humans, decision-related activity is associated with activity in a slightly more dorsomedial vmPFC region^{12,13,18-20}, possibly areas 14m and 11m³⁵⁻³⁷. Nevertheless, in both humans and macaques the activity manifests in subdivisions of area 14 and 11m. Neurons with value- and decision-related activity have been found in a more posterior part of this region⁷¹ although recordings of neural activity made this far rostral in vmPFC have not been reported.

Value-related activity and decision difficulty-related activity in macaque vmPFC

Even though decision-making may be associated with activation changes with different signs in humans and macaques, vmPFC activity may reflect value comparison in both species. If this is the case then, as in humans, we should be able to identify activity in macaque vmPFC covarying with the decision variable guiding choices – the difference between the value of the choice taken as opposed to the choice forgone (contrast of chosen value–unchosen value)¹⁸⁻²⁰. To pursue this question (Supplemental Experimental Procedures; GLM-2), we first performed an initial analysis in which we contrasted trials when a CS+ was chosen with the small number of trials when the no-reward blank screen was chosen. **This analysis allows us to identify activity that is related to choosing any stimulus with any reward association (any CS+) as opposed to any stimulus with no reward association (any CS-) but it does not reveal whether activity tracks the value of the choices.** We found a relative increase in posterior vmPFC (and OFC activity; Fig. S5). Therefore, as suggested by rodent recordings^{72,73} some activity in these regions may not reflect the precise value of a choice but simply whether a choice is guided by stimulus-reward associations.

This initial analysis identified activity linked to the use of stimulus-reward associations *per se* to guide behavior but next we conducted a further analysis focusing only on trials where stimulus-reward associations were being used (we examined just trials on which CS+s were chosen as opposed to those on which a CS with no reward association was chosen). **In this analysis we can identify activity that is related to the specific subjective values of the choices**

that are being considered. By focusing just on trials on which CS+s were chosen we can ensure that the analysis approach we take does not simply identify activity that is related to using any CS+ as opposed to CS- to guide decisions as in the preceding analysis. To perform such an analysis, we estimated the values that choices held for each macaque by measuring the frequency with which they were taken when offered against the other CS+s or unrewarded stimuli (Supplemental Experimental Procedures section: *Parametric value analysis – experiment 2*; Fig. 3c). This resulted in the choice value estimates for each animal plotted in Fig. 3ci which were then used to construct regressors coding for the difference in value between choice taken and rejected plotted in Fig. 3cii. These were then used in a parametric GLM analysis (Experimental Procedures; Table S3: GLM-2, contrast 2). We searched for value difference-related activity in cortex within a 25mm radius of decision-related activity (see Fig. 3a, g). Again, we identified vmPFC activation when brain activity was regressed onto chosen value-unchosen value difference (Fig. 3d; cluster corrected $z > 2.3$; $p < 0.05$). This result suggests that, as in humans, macaque vmPFC activity reflects the decision variable that should guide behavior: the difference between the value of the options chosen and rejected.

The relationship between vmPFC activity and chosen-unchosen value was, however, opposite to that seen in humans; as the difference between values decreased (and so decisions became harder), vmPFC activity increased. We took care to search for a value difference effect like that seen in humans but were unable to find one in macaque vmPFC. This remained the case even if we examined smaller volumes of interest surrounding the peak activation effect associated with decision-making or when we examined the region corresponding to the location of the lesion (and adjacent cortex) studied in experiment 1 (Fig. 3g).

This conclusion was supported by further analyses of parametric BOLD activity changes over time (Fig. 3e, f). In a region of interest (ROI), we extracted the raw BOLD time courses, up sampled and aligned them to the decision onset. For each time point and across trials, we calculated the regression coefficient (“effect size”) associated with chosen-unchosen value difference from the same GLM (Experimental Procedures; Table S3: GLM-2, contrast 2; Fig. 3e). Next we illustrated the impact on vmPFC of parametric increases in the chosen value and the unchosen value separately to confirm that they were negative and positive as expected (Fig. 3f). The impact of the value of the selected option is summarized by a single set of regression coefficients (Fig. 3f, red line); vmPFC activity decreases as the chosen option’s value increases. By contrast, another single set of regression coefficients illustrates how vmPFC activity increases as the value of the unchosen option increases (Fig. 3f, green line). In combination, these two effects mean that vmPFC activity decreases as the difference between choice values (chosen value-unchosen value) increases (as the decision gets easier; Fig. 4).

In summary, the main effect of taking a decision (activity that changes whenever a decision is taken regardless of the values of the options considered; fig.3a,b) is prominent in vmPFC as is activity related to the key variable – the difference in value between the choice taken and the choice rejected – that should drive each decision (fig.3d-f). Activity related to the main effect of decision-making and the decision variable is found in partially overlapping voxels at the statistical threshold level we used. The overlap would be slightly more extensive at a lower threshold. In line with this an analysis conducted in a 25mm radius region of interest (ROI) centered on the peak effect of decision-making (Fig. 3a, g) revealed a significant effect of the key decision variable – the difference in value between the chosen-unchosen options (Fig. S5a).

Additional statistical tests were also used to test the conclusion that vmPFC reflected the key decision variable: the difference in value between the option chosen and rejected. We checked that vmPFC activity still reflected the chosen-unchosen option value difference even when RT (itself also partly determined by the difference in option values) was included in the GLM to explain vmPFC activity. For this analysis, we used appropriate leave-one-out

techniques for both selecting the ROIs and determining the time course peaks ($t_3=-7.5868$, $p=0.0048$).

One possibility is that the vmPFC activity pattern reflects some unusual feature of the CV option in which the subjective value was dissociated from the objective value. In order to test whether this is the case we took three additional measures. Most importantly, we carried out an additional experiment (experiment 3) that we describe below that eschewed the use of a CV option. In addition, however, we carried out additional analyses identical to those shown in Fig. 3e-f but we only included data from trials on which the CV option had not been available. Once again we saw a very similar pattern of activity related to the value of the choice taken and the value of the choice rejected (Fig. S5b).

Another way to check that the interpretation of the vmPFC activity identified by the various parametric GLM analyses described above is not unduly affected by the presence of the CV option is to examine activity related to the presence of each of the choice options (HV, CV, LV, and Unrew) in a complementary analysis (Experimental Procedures; Table S3: GLM-1, contrasts based on all cue-onset regressors and sorted by choice taken; Fig. 4). Because the analyses shown in Fig. 3d-f already suggest that the manner in which the presence of the HV, CV, LV, or Unrew option affects vmPFC activity depends on whether or not it is chosen we cannot look simply at trials containing the HV, CV, LV, or Unrew options; in addition we must consider the context in which each option was presented (what was the other option presented). This complementary analysis had the advantage that it did not depend on precisely how values were assigned to each choice because we simply looked at activity on each of the main decision types. To perform the analysis we took the peak activity from each trial within a 5sec period between 1.5 s and 6.5 s after stimulus presentation (Fig. 4d). It revealed that vmPFC activity reflected the subjective values, rather than the objective reward amounts associated with the stimuli. **This was true regardless of whether the options were simple options such as HV and LV or compound options such as CV.** Moreover, vmPFC increased as the subjective value of the chosen option decreased (compare green and red lines in panels moving left to right; Fig. 4a: HV choices; Fig. 4b: CV choices) and increased as the subjective value of the unchosen option increased (compare red, green, and blue lines; Fig. 4a: HV choices; 4b: CV choices); in a two (chosen value: CV, HV) by two (unchosen value: Unrewarded, LV) factorial ANOVA there was an effect of chosen value ($F_{1,3}=15.236$, $p=0.03$; **after square-root transformation: $F_{1,3}=36.482$, $p=0.009$**) and unchosen value ($F_{1,3}=10.375$, $p=0.049$; **after square-root transformation: $F_{1,3}=21.740$, $p=0.019$**). Such a pattern suggests greater aggregate vmPFC activity when identifying the better option was difficult because it had a low value or because the alternative option had a high value and inspection of the CV-related activation patterns confirms that it is the subjective value of choices, rather than objective reward amount that is correlated with vmPFC activity.

Such a pattern of decision-making behavior and vmPFC activity is consistent with decision-making being mediated by a competition between different pools of neurons that reflect the value of two available options³². The difference in value between two options affects the temporal dynamics with which decision processes can take place in a biophysically plausible way⁴⁷. The observed patterns of aggregate BOLD activity are therefore consistent with the neural dynamics of cell assemblies moving back and forth between different states towards a choice³².

Finally, we note a further analysis of the decisions in experiment 2. We contrasted the decisions that were particularly affected by the vmPFC/OFC lesions in experiment 1 as opposed to the decisions that were less affected by the lesion (GLM1; Fig. S6).

Second, we have tried to follow the reviewer's suggestion and we now show the relationship between the time course peak effects of the different regressors in Fig. 4d (HV, CV, LV, and Unrew). These are shown in the figure as bar plots. Note, however, that the fMRI data

presented in Fig. 4 suggests that the parameter estimates for each option (HV, CV, LV, and Unrew) will only change depending on which other option (again HV, CV, LV, and Unrew) is presented on a given trial because this will determine whether the option is likely to be chosen or rejected. Thus, it only makes sense to use this approach for HV, CV, LV, and unrewarded options as a function of the other option that was available on each trial. This is what we have now done in Fig. 4d in the revised manuscript. Note that the other panels in Fig. 4 relate to exactly the same analysis but in addition they show how the effect on vmPFC activity changes with time.

The revised Fig. 4 now looks like this:

Figure 4. Time courses of vmPFC activity when HV, CV, LV, and unrewarded options are present (ROI from Fig. 3d). These are the four options whose values are illustrated in Fig. 3ci. Because the impact that the option has on vmPFC activity changes depending on what other option is presented on any given trial (and therefore which option is likely to be taken and which is likely to be rejected) the time courses have been sorted by the value of the chosen option: HV (a), CV (b) or LV (c). The effect of the value of the unchosen option can, for example, be seen in panel (a): activity associated with choosing HV is greater when decisions are difficult and choices are made between it and CV (blue) as opposed to LV (green) or Unrewarded (red). The effect of the chosen value can be seen by comparing either the red lines or the green lines in

panels (a) and (b): activity associated with choosing an option increases when it is harder to make the choice because its value is lower. While panels a-c show the time courses of the effects of the various options on vmPFC activity, panel (d) illustrates the same information but using the peaks of the time courses.

The text that refers to Fig. 4 now reads as follows in the revised manuscript:

Another way to check that the interpretation of the vmPFC activity identified by the various parametric GLM analyses described above is not unduly affected by the presence of the CV option is to examine activity related to the presence of each of the choice options (HV, CV, LV, and Unrew) in a complementary analysis (Experimental Procedures; Table S3: GLM-1, contrasts based on all cue-onset regressors and sorted by choice taken; Fig. 4). Because the analyses shown in Fig. 3d-f already suggest that the manner in which the presence of the HV, CV, LV, or Unrew option affects vmPFC activity depends on whether or not it is chosen we cannot look simply at trials containing the HV, CV, LV, or Unrew options; in addition we must consider the context in which it was presented (what was the other option presented). This complementary analysis had the advantage that it did not depend on precisely how values were assigned to each choice because we simply looked at activity on each of the main decision types. To perform the analysis we took the peak activity from each trial within a 5sec period between 1.5 s and 6.5 s after stimulus presentation (Fig. 4d). It revealed that vmPFC activity reflected the subjective values, rather than the objective reward amounts associated with the stimuli. This was true regardless of whether the options were simple options such as HV and LV or compound options such as CV. Moreover, vmPFC increased as the subjective value of the chosen option decreased (compare green and red lines in panels moving left to right; Fig. 4a: HV choices; Fig. 4b: CV choices) and increased as the subjective value of the unchosen option increased (compare red, green, and blue lines; Fig. 4a: HV choices; 4b: CV choices); in a two (chosen value: CV, HV) by two (unchosen value: Unrewarded, LV) factorial ANOVA there was an effect of chosen value ($F_{1,3}=15.236$, $p=0.03$; after square-root transformation: $F_{1,3}=36.482$, $p=0.009$) and unchosen value ($F_{1,3}=10.375$, $p=0.049$; after square-root transformation: $F_{1,3}=21.740$, $p=0.019$). Such a pattern suggests greater aggregate vmPFC activity when identifying the better option was difficult because it had a low value or because the alternative option had a high value and inspection of the CV-related activation patterns confirms that it is the subjective value of choices, rather than objective reward amount that is correlated with vmPFC activity.

5. Also, the authors use a faster HRF which has previously been used for macaque fMRI data analysis. However, the figures reveal that many effects are in fact driven by slower responses. In particular, key results focusing on chosen-unchosen value show a (negative) peak around 8 s (Figure 3e and f). How do the authors explain these slow responses and could the choice of the canonical HRF used for the GLM analysis substantially affect any of the voxel-wise results?

The relatively fast HRF that we used was based on previously reported results of Chau and colleagues (Neuron, 2015). The HRF is also broadly in line with all of the effects in the present study bar one. For example, it is broadly in line with the main effect of the decision event (Fig. 3b), the main effect of the unchosen option in experiment 2 (green line, Fig. 3f), the effect of each option type on vmPFC activity in experiment 2 (Fig. 4), and the effects reported in experiment 3 (Fig. 5b-c). There is just one effect that is more protracted and this is the effect of the chosen option in experiment 2 (red line, Fig. 3f). As a result the chosen-unchosen effect becomes more protracted in Fig. 3e but note that this is only due to the protracted nature of one

component part – the chosen value effect – while the unchosen value effect is transient. We do not know why the chosen value effect was more protracted in experiment 2 and are hesitant about over-interpreting it in experiment 3. It may, however, relate to a reactivation of the chosen value at the time that the outcome is delivered several seconds later (Akaishi et al., 2016) and the fact that the delay between decision and outcome was longer in experiment 2 than experiment 3. In the revised manuscript we have simply noted that the results remained similar when the analyses were conducted with longer HRFs as follows:

We noticed that the effect of the chosen value in experiment 2 (although not experiment 3) was protracted and it is possible that this may be related to the reactivation of the representation of the chosen option at the time of outcome delivery⁷⁴ and the fact that the decision and outcome separation was longer in experiment 2 than experiment 3. Further analyses using slower HRFs (with a hemodynamic lag of 4sec) failed to identify additional areas of activity.

References

1. Boorman, E. D., Behrens, T. E. J., Woolrich, M. W. & Rushworth, M. F. S. How green is the grass on the other side? Frontopolar cortex and the evidence in favor of alternative courses of action. *Neuron* **62**, 733–743 (2009).
2. Chau, B. K. H., Kolling, N., Hunt, L. T., Walton, M. E. & Rushworth, M. F. S. A neural mechanism underlying failure of optimal choice with multiple alternatives. *Nat. Neurosci.* **17**, 463–470 (2014).
3. De Martino, B., Fleming, S. M., Garrett, N. & Dolan, R. J. Confidence in value-based choice. *Nat. Neurosci.* **16**, 105–110 (2013).
4. Hunt, L. T. *et al.* Mechanisms underlying cortical activity during value-guided choice. *Nat. Neurosci.* **15**, 470–476, S1-3 (2012).
5. Jocham, G., Hunt, L. T., Near, J. & Behrens, T. E. J. A mechanism for value-guided choice based on the excitation-inhibition balance in prefrontal cortex. *Nat. Neurosci.* **15**, 960–961 (2012).
6. Jocham, G. *et al.* Dissociable contributions of ventromedial prefrontal and posterior parietal cortex to value-guided choice. *NeuroImage* **100**, 498–506 (2014).
7. Philiastides, M. G., Biele, G. & Heekeren, H. R. A mechanistic account of value computation in the human brain. *Proc. Natl. Acad. Sci. U. S. A.* **107**, 9430–9435 (2010).
8. Wunderlich, K., Dayan, P. & Dolan, R. J. Mapping value based planning and extensively trained choice in the human brain. *Nat. Neurosci.* **15**, 786–791 (2012).
9. Mackey, S. & Petrides, M. Quantitative demonstration of comparable architectonic areas within the ventromedial and lateral orbital frontal cortex in the human and the macaque monkey brains. *Eur. J. Neurosci.* **32**, 1940–1950 (2010).
10. Mackey, S. & Petrides, M. Architecture and morphology of the human ventromedial prefrontal cortex. *Eur. J. Neurosci.* **40**, 2777–2796 (2014).

11. Kable, J. W. & Glimcher, P. W. The neural correlates of subjective value during intertemporal choice. *Nat. Neurosci.* **10**, 1625–1633 (2007).
12. Abitbol, R. *et al.* Neural mechanisms underlying contextual dependency of subjective values: converging evidence from monkeys and humans. *J. Neurosci. Off. J. Soc. Neurosci.* **35**, 2308–2320 (2015).
13. Howard, J. D., Gottfried, J. A., Tobler, P. N. & Kahnt, T. Identity-specific coding of future rewards in the human orbitofrontal cortex. *Proc. Natl. Acad. Sci. U. S. A.* **112**, 5195–5200 (2015).
14. Lebreton, M., Abitbol, R., Daunizeau, J. & Pessiglione, M. Automatic integration of confidence in the brain valuation signal. *Nat. Neurosci.* **18**, 1159–1167 (2015).
15. Li, Y., Vanni-Mercier, G., Isnard, J., Mauguière, F. & Dreher, J.-C. The neural dynamics of reward value and risk coding in the human orbitofrontal cortex. *Brain J. Neurol.* **139**, 1295–1309 (2016).
16. Noonan, M. P., Mars, R. B. & Rushworth, M. F. S. Distinct roles of three frontal cortical areas in reward-guided behavior. *J. Neurosci. Off. J. Soc. Neurosci.* **31**, 14399–14412 (2011).
17. Lim, S.-L., O'Doherty, J. P. & Rangel, A. The decision value computations in the vmPFC and striatum use a relative value code that is guided by visual attention. *J. Neurosci. Off. J. Soc. Neurosci.* **31**, 13214–13223 (2011).
18. Philiastides, M. G., Biele, G. & Heekeren, H. R. A mechanistic account of value computation in the human brain. *Proc. Natl. Acad. Sci. U. S. A.* **107**, 9430–9435 (2010).
19. Hunt, L. T. *et al.* Mechanisms underlying cortical activity during value-guided choice. *Nat. Neurosci.* **15**, 470–476, S1-3 (2012).
20. Chau, B. K. H., Kolling, N., Hunt, L. T., Walton, M. E. & Rushworth, M. F. S. A neural mechanism underlying failure of optimal choice with multiple alternatives. *Nat. Neurosci.* **17**, 463–470 (2014).

21. Boorman, E. D., Behrens, T. E. J., Woolrich, M. W. & Rushworth, M. F. S. How green is the grass on the other side? Frontopolar cortex and the evidence in favor of alternative courses of action. *Neuron* **62**, 733–743 (2009).
22. Jocham, G., Hunt, L. T., Near, J. & Behrens, T. E. J. A mechanism for value-guided choice based on the excitation-inhibition balance in prefrontal cortex. *Nat. Neurosci.* **15**, 960–961 (2012).
23. Jocham, G. *et al.* Dissociable contributions of ventromedial prefrontal and posterior parietal cortex to value-guided choice. *NeuroImage* **100**, 498–506 (2014).
24. Kolling, N., Behrens, T. E. J., Mars, R. B. & Rushworth, M. F. S. Neural mechanisms of foraging. *Science* **336**, 95–98 (2012).
25. Wunderlich, K., Dayan, P. & Dolan, R. J. Mapping value based planning and extensively trained choice in the human brain. *Nat. Neurosci.* **15**, 786–791 (2012).
26. De Martino, B., Fleming, S. M., Garrett, N. & Dolan, R. J. Confidence in value-based choice. *Nat. Neurosci.* **16**, 105–110 (2013).
27. Kolling, N., Wittmann, M. & Rushworth, M. F. S. Multiple neural mechanisms of decision making and their competition under changing risk pressure. *Neuron* **81**, 1190–1202 (2014).
28. Lopez-Persem, A., Domenech, P. & Pessiglione, M. How prior preferences determine decision-making frames and biases in the human brain. *eLife* **5**, (2016).
29. Vanduffel, W., Zhu, Q. & Orban, G. A. Monkey cortex through fMRI glasses. *Neuron* **83**, 533–550 (2014).
30. Wilson, B. *et al.* Auditory sequence processing reveals evolutionarily conserved regions of frontal cortex in macaques and humans. *Nat. Commun.* **6**, 8901 (2015).
31. Wallis, J. D. Cross-species studies of orbitofrontal cortex and value-based decision-making. *Nat. Neurosci.* **15**, 13–19 (2011).
32. Rich, E. L. & Wallis, J. D. Decoding subjective decisions from orbitofrontal cortex. *Nat. Neurosci.* **19**, 973–980 (2016).

33. Rudebeck, P. H. & Murray, E. A. Dissociable effects of subtotal lesions within the macaque orbital prefrontal cortex on reward-guided behavior. *J. Neurosci. Off. J. Soc. Neurosci.* **31**, 10569–10578 (2011).
34. Rudebeck, P. H., Saunders, R. C., Prescott, A. T., Chau, L. S. & Murray, E. A. Prefrontal mechanisms of behavioral flexibility, emotion regulation and value updating. *Nat. Neurosci.* **16**, 1140–1145 (2013).
35. Mackey, S. & Petrides, M. Architecture and morphology of the human ventromedial prefrontal cortex. *Eur. J. Neurosci.* **40**, 2777–2796 (2014).
36. Mackey, S. & Petrides, M. Quantitative demonstration of comparable architectonic areas within the ventromedial and lateral orbital frontal cortex in the human and the macaque monkey brains. *Eur. J. Neurosci.* **32**, 1940–1950 (2010).
37. Neubert, F.-X., Mars, R. B., Sallet, J. & Rushworth, M. F. S. Connectivity reveals relationship of brain areas for reward-guided learning and decision making in human and monkey frontal cortex. *Proc. Natl. Acad. Sci. U. S. A.* **112**, E2695-2704 (2015).
38. Stephens, D. W. & Krebs, J. R. *Foraging theory*. (Princeton University Press, 1986).
39. List, J. A. Preference Reversals of a Different Kind: The ‘More Is Less’ Phenomenon. *Am. Econ. Rev.* **92**, 1636–1643 (2002).
40. Hsee, C. K. Less is better: when low-value options are valued more highly than high-value options. *J. Behav. Decis. Mak.* **11**, 107–121 (1998).
41. Kralik, J. D., Xu, E. R., Knight, E. J., Khan, S. A. & Levine, W. J. When less is more: evolutionary origins of the affect heuristic. *PloS One* **7**, e46240 (2012).
42. Fellows, L. K. & Farah, M. J. The role of ventromedial prefrontal cortex in decision making: judgment under uncertainty or judgment per se? *Cereb. Cortex N. Y. N* **17**, 2669–2674 (2007).
43. Pearson, J. M., Watson, K. K. & Platt, M. L. Decision making: the neuroethological turn. *Neuron* **82**, 950–965 (2014).

44. Kolling, N. *et al.* Value, search, persistence and model updating in anterior cingulate cortex. *Nat. Neurosci.* **19**, 1280–1285 (2016).
45. Hämmerer, D., Bonaiuto, J., Klein-Flügge, M., Bikson, M. & Bestmann, S. Selective alteration of human value decisions with medial frontal tDCS is predicted by changes in attractor dynamics. *Sci. Rep.* **6**, 25160 (2016).
46. Hare, T. A., Schultz, W., Camerer, C. F., O'Doherty, J. P. & Rangel, A. Transformation of stimulus value signals into motor commands during simple choice. *Proc. Natl. Acad. Sci. U. S. A.* **108**, 18120–18125 (2011).
47. Wang, X.-J. Probabilistic decision making by slow reverberation in cortical circuits. *Neuron* **36**, 955–968 (2002).
48. Wong, K.-F. & Wang, X.-J. A recurrent network mechanism of time integration in perceptual decisions. *J. Neurosci. Off. J. Soc. Neurosci.* **26**, 1314–1328 (2006).
49. Sescousse, G., Redouté, J. & Dreher, J.-C. The architecture of reward value coding in the human orbitofrontal cortex. *J. Neurosci. Off. J. Soc. Neurosci.* **30**, 13095–13104 (2010).
50. Crittenden, B. M., Mitchell, D. J. & Duncan, J. Recruitment of the default mode network during a demanding act of executive control. *eLife* **4**, e06481 (2015).
51. Kolling, N., Behrens, T., Wittmann, M. K. & Rushworth, M. Multiple signals in anterior cingulate cortex. *Curr. Opin. Neurobiol.* **37**, 36–43 (2016).
52. Louie, K., Grattan, L. E. & Glimcher, P. W. Reward value-based gain control: divisive normalization in parietal cortex. *J. Neurosci. Off. J. Soc. Neurosci.* **31**, 10627–10639 (2011).
53. Louie, K., Khaw, M. W. & Glimcher, P. W. Normalization is a general neural mechanism for context-dependent decision making. *Proc. Natl. Acad. Sci. U. S. A.* **110**, 6139–6144 (2013).
54. Louie, K., Glimcher, P. W. & Webb, R. Adaptive neural coding: from biological to behavioral decision-making. *Curr. Opin. Behav. Sci.* **5**, 91–99 (2015).

55. Louie, K., LoFaro, T., Webb, R. & Glimcher, P. W. Dynamic divisive normalization predicts time-varying value coding in decision-related circuits. *J. Neurosci. Off. J. Soc. Neurosci.* **34**, 16046–16057 (2014).
56. Soltani, A., De Martino, B. & Camerer, C. A range-normalization model of context-dependent choice: a new model and evidence. *PLoS Comput. Biol.* **8**, e1002607 (2012).
57. Murray, E. A., Moylan, E. J., Saleem, K. S., Basile, B. M. & Turchi, J. Specialized areas for value updating and goal selection in the primate orbitofrontal cortex. *eLife* **4**, (2015).
58. Baxter, M. G. & Murray, E. A. Impairments in visual discrimination learning and recognition memory produced by neurotoxic lesions of rhinal cortex in rhesus monkeys. *Eur. J. Neurosci.* **13**, 1228–1238 (2001).
59. Hikosaka, O., Kim, H. F., Yasuda, M. & Yamamoto, S. Basal ganglia circuits for reward value-guided behavior. *Annu. Rev. Neurosci.* **37**, 289–306 (2014).
60. Yamamoto, S., Kim, H. F. & Hikosaka, O. Reward value-contingent changes of visual responses in the primate caudate tail associated with a visuomotor skill. *J. Neurosci. Off. J. Soc. Neurosci.* **33**, 11227–11238 (2013).
61. Passingham, R. E. *The frontal lobes and voluntary action / R.E. Passingham.* (Oxford University Press, 1993).
62. Rich, E. L. & Wallis, J. D. Medial-lateral organization of the orbitofrontal cortex. *J. Cogn. Neurosci.* **26**, 1347–1362 (2014).
63. Boutelet, S. & Richmond, B. J. Ventromedial and orbital prefrontal neurons differentially encode internally and externally driven motivational values in monkeys. *J. Neurosci. Off. J. Soc. Neurosci.* **30**, 8591–8601 (2010).
64. Cai, X. & Padoa-Schioppa, C. Contributions of orbitofrontal and lateral prefrontal cortices to economic choice and the good-to-action transformation. *Neuron* **81**, 1140–1151 (2014).
65. Jones, J. L. *et al.* Orbitofrontal cortex supports behavior and learning using inferred but not cached values. *Science* **338**, 953–956 (2012).

66. Rudebeck, P. H. & Murray, E. A. The orbitofrontal oracle: cortical mechanisms for the prediction and evaluation of specific behavioral outcomes. *Neuron* **84**, 1143–1156 (2014).
67. Stalnaker, T. A., Cooch, N. K. & Schoenbaum, G. What the orbitofrontal cortex does not do. *Nat. Neurosci.* **18**, 620–627 (2015).
68. Schoenbaum, G. & Roesch, M. Orbitofrontal cortex, associative learning, and expectancies. *Neuron* **47**, 633–636 (2005).
69. Málková, L., Gaffan, D. & Murray, E. A. Excitotoxic lesions of the amygdala fail to produce impairment in visual learning for auditory secondary reinforcement but interfere with reinforcer devaluation effects in rhesus monkeys. *J. Neurosci. Off. J. Soc. Neurosci.* **17**, 6011–6020 (1997).
70. Noonan, M. P. *et al.* Separate value comparison and learning mechanisms in macaque medial and lateral orbitofrontal cortex. *Proc. Natl. Acad. Sci. U. S. A.* **107**, 20547–20552 (2010).
71. Strait, C. E., Blanchard, T. C. & Hayden, B. Y. Reward value comparison via mutual inhibition in ventromedial prefrontal cortex. *Neuron* **82**, 1357–1366 (2014).
72. Lopatina, N. *et al.* Lateral orbitofrontal neurons acquire responses to upshifted, downshifted, or blocked cues during unblocking. *eLife* **4**, e11299 (2015).
73. Lopatina, N. *et al.* Medial Orbitofrontal Neurons Preferentially Signal Cues Predicting Changes in Reward during Unblocking. *J. Neurosci. Off. J. Soc. Neurosci.* **36**, 8416–8424 (2016).
74. Akaishi, R., Kolling, N., Brown, J. W. & Rushworth, M. Neural Mechanisms of Credit Assignment in a Multicue Environment. *J. Neurosci. Off. J. Soc. Neurosci.* **36**, 1096–1112 (2016).

REVIEWERS' COMMENTS:

Reviewer #1 (Remarks to the Author):

The authors have largely addressed most of my original concerns. I only have one remaining suggestion pertaining to comment #5 in my original review (signed vs unsigned value difference analysis). The authors argue that the signed difference has a significant impact on vmPFC while the absolute value does not. In turn, this is interpreted as evidence that "vmPFC activity is intimately related to the guidance of behavior".

I agree with this statement, however I think it would be useful for the reader if the authors unpacked this further to explain how such *signed* quantities might be represented in the brain and used to influence choice behavior.

For example would the brain interpret large increases and decreases in vmPFC as evidence supporting each of the two alternatives respectively or would it perhaps represent the probability of choosing one of the two items (e.g. probability of choosing the higher valued item). Though both of these interpretations could ultimately lead to the same outcome, computationally they represent different mechanisms.

Reviewer #2 (Remarks to the Author):

The authors have done an excellent work in addressing my concerns. I believe that the manuscript is now acceptable for publication.

Reviewer #3 (Remarks to the Author):

The authors have adequately addressed all of my initial comments. I have no further comments.

Reviewers' comments:

Reviewer #1 (Remarks to the Author):

The authors have largely addressed most of my original concerns. I only have one remaining suggestion pertaining to comment #5 in my original review (signed vs unsigned value difference analysis). The authors argue that the signed difference has a significant impact on vmPFC while the absolute value does not. In turn, this is interpreted as evidence that “vmPFC activity is intimately related to the guidance of behavior”.

I agree with this statement, however I think it would be useful for the reader if the authors unpacked this further to explain how such *signed* quantities might be represented in the brain and used to influence choice behavior.

For example would the brain interpret large increases and decreases in vmPFC as evidence supporting each of the two alternatives respectively or would it perhaps represent the probability of choosing one of the two items (e.g. probability of choosing the higher valued item). Though both of these interpretations could ultimately lead to the same outcome, computationally they represent different mechanisms.

This is a very good comment. Some previous work with high temporal resolution recordings in human MEG and single unit recording studies, such as those of Hunt and colleagues⁹ and Rich and Wallis²², attempted to address this issue. Essentially these studies suggest that there is a transition between the two coding schemes that the reviewer considers. The transition occurs between the time of the decision process itself and the end of the decision process when the choice is taken. During the decision process the activity reflects the relative evidence in favor of one option rather than another but by the end of the decision the activity may just reflect features of the choice that is going to be taken. We have noted this in the revised manuscript as follows:

High temporal resolution recording studies in humans⁹ and macaques²² suggest that there is a transition between vmPFC/OFC activity reflecting relative evidence in favor of one option rather than another during the decision period to activity just reflecting the value of the choice that is taken by the end of the decision.

References

9. Hunt, L. T. *et al.* Mechanisms underlying cortical activity during value-guided choice. *Nat. Neurosci.* **15**, 470–476, S1-3 (2012).
22. Rich, E. L. & Wallis, J. D. Decoding subjective decisions from orbitofrontal cortex. *Nat. Neurosci.* **19**, 973–980 (2016).